

# Evaposublimation from the snow in the Mediterranean mountains of Sierra Nevada (Spain)

Javier Herrero[1] and María José Polo[2]

[1]Fluvial Dynamics and Hydrology Research Group. Andalusian Institute for Earth System Research. University of Granada. Avda. del Mediterráneo s/n 18006, Granada, Spain
[2]Fluvial Dynamics and Hydrology Research Group. Andalusian Institute for Earth System Research. University of Córdoba. Rabanales Campus, Leonardo da Vinci Building 14071, Córdoba, Spain.

*Correspondence to:* Javier Herrero (herrero@ugr.es)

**Abstract.** In this study we quantify the evaposublimation and the energy balance of the seasonal snowpack in the Mediterranean semiarid region of Sierra Nevada, Spain (37° N). In these kinds of regions, the incidence of this return of water to the atmosphere is particularly significant to the hydrology and water availability. The analysis of the evaposublimation from snow allows us to deduct the losses of water expected in the short and medium term, and is critical for the efficient planning of this

basic and scarce resource. To achieve this, we performed 15 test field campaigns from 2009 to 2015, during which detailed measurements of mass fluxes of a controlled volume of snow were recorded using a modified version of an evaporation pan with lysimeter. Meteorological data at the site of the snow control volume was extensively monitored during the tests. With these data, a point energy balance snowmelt model was validated for the area. This model, fed with the complete meteorological dataset available at the Refugio Poqueira Station (2500 m.a.s.l.), let us estimate that evaposublimation losses for this site

can range from 24 to 33% of total annual ablation. This ratio is changeable throughout the year and between years, depending on the particular combination and timing of the meteorological inputs, generally unforeseeable in this semiarid region. Evaposublimation proceeds at maximum rates of up to 0.49 $mm\,h^{-1}$, an order of magnitude less than maximum melt rates. However, evaposublimation occurs during 60% of the time that snow lies, while snowmelt only takes up 10% of this time. Hence, both processes remain close in magnitude on the annual scale.

## 1    Introduction

Seasonal snow can occur in temperate areas at increasing altitudes as the latitude descends. In these mountainous regions, snow becomes the primary source of water during the year (Shaban et al., 2004) and rules its availability and timing. Snow plays a vital role as a source of water supply for human consumption, irrigation, and survival of species and habitats during the dry season. Any debate and management decision regarding water use and sustainability in these drought–prone areas

must be based on the accurate knowledge of the snowpack dynamics. In this context, the partitioning of ablation into melting and evaporation/sublimation determines the water return to the atmosphere and the replenishment of surface and groundwater. This is particularly relevant in a scenario of global warming that implies a potential snow regression in these areas because of impacts on the energy and mass flux regimes (Pérez-Palazón et al., 2015).





Significant research has been carried out on snow dynamics (Garstka, 1964; Mellor, 1964; Colbeck, 1982; Morris, 1989), especially on the description of the energy balance that drives the different mass fluxes of ablation that affect the snowpack (Anderson, 1968; Kuusisto, 1986; Jordan, 1991; Marks and Dozier, 1992; Tarboton and Luce, 1996). Generalization for mountainous areas is particularly difficult as energy balance changes with elevation, aspect and vegetation cover since these factors

modify the local temperature, wind exposure, and shadowing of solar and longwave radiation. Besides, in Mediterranean regions these meteorological variables are subject to the characteristic irregular weather patterns. As a consequence of this variability, annual snowmelt timing can shift from a single typical main springtime melting cycle to several mid–winter partial or complete melting cycles.

The snow dynamics in semiarid environments is so dependent on the energy state of the snowpack, that accurate modelling

usually requires physical approaches that calculate the energy balance (Schulz and de Jong, 2004). Many of the problems usually found when validating the models and quantifying the actual evaporation taking place are due to the difficulty of taking measurements under rough winter conditions typical of high mountain areas. To begin with, automatic ground sensors are difficult to maintain operational long enough to obtain significant data series over a period of years. In addition, the spatial variability of the snowpack makes it difficult to produce a realistic estimate of the snow processes, which change substantially

over small distances, according to aspect and elevation. Satellite images are a good source of distributed data, but they mostly provide direct information only about the presence or absence of snow (Hall et al., 2002). Research is currently being carried out into the estimation of other snow variables, like snow water equivalent, from satellite sources, and in most cases it requires the joint use of remote sensing and energy balance modelling (Cline et al., 1998; Molotch and Margulis, 2008).

One of the mass balance fluxes of snow is the water vapour exchange between the snow surface and the atmosphere, and

is directly linked to the latent heat balance. Evaporation and sublimation of water from the snow surface occur alternately depending on the phase it is in. The evaposublimation rate depends on the vapour pressure gradient between the surface of the snow and the air, which is mainly influenced by the local wind intensity, and hence, by the complex turbulent phenomena occurring in the boundary layer. This makes both its measurement and its simulation one of the most complicated elements of all the fluxes involved in the energy balance in the snowpack. Numerous studies have focused on measuring and estimating

evaposublimation losses from snowpacks in forested (Schmidt et al., 1998; Molotch et al., 2007) and unforested areas (Pomeroy and Essery, 1999; Fassnacht, 2004). Evaposublimation rates are substantially enhanced in the latter (West, 1962). Mountainous areas provide particularly good conditions for evaposublimation due to their inherently lower vapour pressure and higher wind speed (Gray and Prowse, 1993). In this kind of topography, it is not uncommon to experience periods with strong wind and low humidity (Herrero et al., 2009) that trigger high evaposublimation rates. Schulz and de Jong (2004) remark that high solar

radiation and rising air temperatures support evaposublimation as long as the snowpack remains cold and snowmelt does not dominate in the ablation process.

Given the conditions described for semiarid mountainous areas, the spatial and temporal variation of the evaposublimation rates from the snow may be considerable. Under this constraint, it is difficult to give a meaningful average value. Leydecker and Melack (2000) calculated that the average annual evaporation from the snowpack was 36% of the annual precipitation in Sierra

Nevada, California (37°N and 3000 m.a.s.l.), while its magnitude varied from 12 to 156 mm between years (Leydecker and



Melack, 1999). Froyland (2013) estimated the amount of evaposublimation in the semiarid San Francisco Peaks of the Colorado plateau (35°N and 2100 m.a.s.l.) in a range of between 17% and 43% of the annual snowfall. A similar result was obtained by Herrero et al. (2009) in Sierra Nevada, Spain (37°N and 2500 m.a.s.l.), where an annual evaposublimation of between 21% and 42% was calculated for two consecutive years using a physical snow model. This change in the evaposublimation rates from the snow raises doubts about its actual effect on the overall basin hydrology. Only the dating of snow accumulation is accepted as a critical factor for establishing the annual evaposublimation rate and the runoff efficiency of high elevation snowpacks (Avery et al., 1992).

Evaposublimation from the snowpack can be measured at single points on the ground using different methodologies. 1) Snow water equivalent sensors (Johnson and Marks, 2004) and snowmelt lysimeters with snowpillows (Tekeli et al., 2005) are used in conjunction with the methodology developed for studying evapotranspiration on agricultural lands. The main problem of these devices is poor correspondence between the meltwater produced at the snow surface and the water arriving at the base of the snowpack on a unit–area basis (Kattelmann, 2000). This is due to several factors such us snow bridging, sensitivity of the sensors to changes in humidity and temperature, and their need for constant maintenance because of the adverse meteorology. Besides, unenclosed snowmelt lysimeters allow lateral flow of water into and out of the column of snow overlying the collector. In semiarid mountain areas, the snow pillow measurements are adversely influenced by the typically shallow snow cover and the frequently high wind speed (Schulz and de Jong, 2004). 2) A second approach is based on eddy covariance (EC) systems (Baldocchi et al., 1988) for direct measurement of the vertical turbulent fluxes of sensible and latent heat from the snowpack. This technology has been employed to calculate sublimation over snow in terrains of varying complexity (Pomeroy and Essery, 1999; Molotch et al., 2007; Marks et al., 2008), most of them in forested areas and over short periods of time. EC systems allow for the most direct measurement of latent heat flux and provide valuable high–resolution (typically 10 Hz) time data series. However, the instruments required for taking these measurements are complex, fragile, and require large, clear, low–angle areas to function optimally (Froyland, 2013). Experiments using EC systems are expensive and time consuming, as the data obtained demand rigorous analysis with corrections and post–processing to ensure measurement accuracy (Reba et al., 2009). 3) A final approach is based on the evaporation pan method (Doty and Johnston, 1969; Föhn, 1973; Lemmelä and Kuusisto, 1974; Avery et al., 1992; Radionov et al., 1997; Hachikubo, 2001). This traditional technology is a simple, inexpensive, and portable means of measurement based on the monitoring of a sample of snow collected *in situ* into a container that does not appreciably alter the natural snow conditions. It can be considered as a small–scale snowmelt lysimeter that works for short periods of time during which the device is not left unattended. This methodology has been commonly used in alpine environments (Kaser, 1982; Suzuki et al., 1999; Jackson and Prowse, 2009; Froyland, 2013) where rough meteorological conditions prevent the use of the more precise but delicate instrumentation.

The objective of this work is to assess the significance and time variability of the evaposublimation losses from the snowpack in the Mediterranean semiarid mountains of Sierra Nevada (Spain). Model performance and reliability is tested against direct measurements of evaposublimation and melting carried out using a portable version of an evaporation pan with lysimeter in 15 field campaigns throughout this region under different weather conditions from 2009 to 2015. Insights into the processes governing evaposublimation are obtained by using an energy balance snow model (Herrero et al., 2009) fed by a detailed me-



teorological dataset (2008–2015) from the Refugio Poqueira weather station (2500 m.a.s.l.), and calibrated using the recorded ablation data.

## 2   Study area

Sierra Nevada is a linear mountain range, 90 km long and 20 km wide, parallel to the Mediterranean coastline of southern Spain and situated at an approximate latitude of 37°N. The highest peak stands at 3479 m.a.s.l. at a distance of approximately 40 km from the sea (Fig. 1). This "island" of high mountain climate and snow surrounded by Mediterranean semiarid conditions is relatively recent in geological terms, having been formed during the Alpine Orogeny, that brought out ancient materials of the Triassic period. Sierra Nevada's more than twenty peaks over 3000 m.a.s.l. are aligned along a west–east axis that divides the area into a north–continental and a south–Mediterranean climatic zone. These zones exhibit strong differences associated with their topographic gradients, coastal exposure to the south, and the prevailing winds from the west. The northern face hosts a major ski resort, the southernmost of Western Europe, which relies upon artificially created snow to maintain a continuous snowpack during the whole ski season, typically from late November to April/May.

Physical characteristics and location have favoured a rich flora and fauna, and have led to Sierra Nevada's classification as a biodiversity hotspot, with official recognition as a UNESCO Biosphere Reserve (1986), National Park (1999), LTER (Long–Term Ecological Research) site (2008), and Nature 2000 site (2012). Snow occurrence and persistence are the main drivers of the hydrological dynamics of Sierra Nevada. By buffering the generation of runoff and maintaining soil moisture, these dynamics prolong water flow in rivers well past the wet season, and ultimately determine habitat distribution.

During the winter season, a continuous snowpack is likely to persist above 2500–3000 m.a.s.l., though often interrupted by periods of intense melting. Even during the summer, areas of patchy snow can be found above 3000 m.a.s.l. in wind–protected spots, especially on the north face, even though their maintenance between years is subject to the intensity and timing of the snowfall events. Precipitation varies greatly in space, with elevation, longitude and face (north/south), and between years. The mean annual precipitation on the west side of the Sierra, facing the prevailing direction of incoming storms, is 550 mm at 1000 m.a.s.l. and 750 mm at 2000 m.a.s.l. On the opposite side, on the east and the north–east, there is an important rain shadow effect that diminishes this mean annual precipitation down to 300 mm at 1000 m.a.s.l. and 465 mm at 2000 m.a.s.l. The mean gradient of precipitation with elevation along the entire Sierra is about 150 mm km$^{-1}$ above 1300 m.a.s.l. Snowfalls occur mainly from November to April at altitudes above 2000 m.a.s.l. At the Refugio Poqueira weather station (2500 m.a.s.l; Fig. 1), the average precipitation is 889 mm per year, 59% of which occurs as snow. The variability between years makes the precipitation oscillate greatly between 1426 mm for a wet year and 520 mm for a dry one. The fraction of solid precipitation also varies between 88% and 46%, with a general tendency to be higher with lower annual precipitation values. The difference in total snowfall varies from 910 mm in a wet year to 335 mm in a dry year. The amount of rain on accumulated snow (rain–on–snow events) averages 117 mm per year, ranging from 7 to 223 mm depending on the particular circumstances of each year. The wind speed is high, with an average of 3.4 m s$^{-1}$ at this station.




To study the evaposublimation dynamics, 15 field surveys were carried out on both the northern and southern faces of Sierra Nevada. The southern tests were performed at the Refugio Poqueira, a monitoring site which has been operational since 2004. It is equipped with an alter–shielded rain gauge, and sensors monitoring the main variables of impact on the energy balance of the snow: temperature, relative humidity, short and longwave radiation (since 2008), wind speed, and pressure. Since 2009, a digital camera has registered the daily spatial variation of the snow cover fraction and depth over a 30x30 m area at this site (Pimentel et al., 2015). Snow surveys are carried out throughout the winter season, with systematic measurements of the physical properties of the snowpack.

Tests on the north side were conducted above the town of Pradollano (Fig. 1), at different locations with elevation ranging between 2400 and 2600 m.a.s.l., since no permanent monitoring sites were available on this face. Portable weather stations were used during the surveys to monitor the weather variables listed above. Although the weather conditions are not very different from those in the south in terms of precipitation and cloudiness, the north–facing slopes receive lower effective surface radiation than south–facing ones. This favours the existence of shady areas, especially in winter, which in turn affect temperature and soil moisture. Snow remains colder and stays on the surface for longer. Vegetation and ecosystems are also affected by this shadow effect (Dionisio et al., 2012).

## 3   Methodology

The energy and mass balance snow model designed by Herrero et al. (2009) was used to analyse the ablation processes at the study area from the field data collected during different surveys performed on the snowpack from 2009 to 2015. During these field surveys, the actual evaposublimation and melt rates were measured under different atmospheric conditions and at different stages of the snow season. The evaposublimation regime was assessed from the model simulation for the complete study period 2008?-2015 using 5–min weather data series available from the Refugio Poqueira monitoring site.

### 3.1   Snow model

The energy and mass balance equations in Herrero et al. (2009) are applied over a 1–layer vertical column in the snowpack to simulate the evolution of the snow water equivalent (SWE), the snow depth, and both the snowmelt and evaposublimation fluxes. The model is driven by time series of the following meteorological data: precipitation, air temperature, relative humidity, wind speed, solar radiation and incoming longwave radiation. Its physical structure follows the approach of models like those presented by Tarboton and Luce (1996) or Koivusalo and Kokkonen (2002). This kind of energy balance models with a simplified snowpack structure have provided a reliable performance and short runtime, while making use of a limited number of parameters to provide an appropriate representation of the processes that govern the snow dynamics (Magnusson et al., 2015). In this model, two calibration parameters are selected (Herrero et al., 2009), the snow roughness and the sensible heat–transfer coefficient under windless conditions.

The basic equations of the balance, as well as the definition of the different mass and energy terms, are included in A, while further details of the numerical resolution of the algorithm and its application to Sierra Nevada can be found in Herrero et al.





(2009). Some improvements on the original model regarding snowfall partition, albedo and longwave radiation are described below.

Precipitation is directly measured by the weather station in every case. It is partitioned into rain or snow using the wet bulb temperature, $T_w$. This is taken as the temperature for the rain, which is considered frozen if $T_w <=0°$C. $T_w$ can be estimated from the temperature of the air, $T_a$, the relative humidity of the air, $W_a$, and the atmospheric pressure, $P_a$, using Normand's Rule (e.g., Stull, 2000).

The shortwave albedo of the snow, $\alpha$, is a property of the snowpack surface that changes with time, usually decreasing as the snow grain size increases. Albedo plays an important role in the energy balance of the snow, especially during the melting periods. This influence is even greater in semiarid mountainous areas because of the high solar radiation rates, small zenith angles and a very dry and clean atmosphere (Aguilar et al., 2010; Abermann et al., 2014). In this model, if no measurements are available, $\alpha$ is parametrised as a function of the snow surface age between a maximum value of 0.8 for recent snow and a minimum of 0.4 for old snow. The ageing process is predicted with a linear decay with time (Baker et al., 1990; Pimentel et al., 2013) slower for cold snow (0.006 day$^{-1}$) than for melting snow (0.018 day$^{-1}$). New snowfall refreshes $\alpha$ at a rate of 0.05 mm$^{-1}$ of new SWE.

The equivalent atmospheric emissivity, $\varepsilon_a$, is used to calculate the incoming longwave radiation, $L_{down}$, from the temperature of the atmosphere. The model can work with direct measurements of $L_{down}$ where available, or estimate $\varepsilon_a$ from the near–surface temperature, relative humidity and clearness index (related to solar radiation and cloudiness) using the empirical expression of Herrero and Polo (2012) .

Finally, the snow model includes two main parameters, explained in A, that are usually subject to calibration under semi-arid/Mediterranean conditions when no reliable and/or extensive measurements are available for them: the aerodynamic roughness length of the snow $z_0$, a parameter with influence on both the bulk latent–heat and sensible–heat transfer coefficients, $K_{UE}$ (Eq. A5) and $K_H$ (Eq. A6), respectively, and the sensible–heat transfer coefficient in windless conditions, $K_{H0}$ (Eq. A4).

Although the concept of aerodynamic roughness length is well defined from a theoretical viewpoint, it is difficult to establish its real value under field conditions (Calanca, 2001). Anderson (1976) measured values for seasonal snow cover that vary from 0.1 to 38 mm, while Dingman (2002) proposed reducing this interval to values between 0.5 and 5 mm. King et al. (2008) show measurements for $z_0$ that lay in the interval from 0.2 to 4 mm for seasonal snow cover, even though these values can increase up to 20 mm on metamorphosed snow with undulating surfaces. This corresponds to a range of two degrees of magnitude for a parameter that has an important effect on the result of the mass and energy balance in Eq. (A1) and Eq. (A2) (Hock, 2005; Brun et al., 2008). Its simulation is complicated and problematic since, as well as evolving over time for the same surface during its metamorphosis (Plüss and Mazzoni, 1994) and being related to wind speed (Andreas, 2011), it lacks a well–defined physical meaning (Hock, 2005). For simplicity, in many modelling applications $z_0$ is regarded as constant throughout the simulation (Tarboton and Luce, 1996; Essery et al., 1999), and no discrimination is made between $z_0$ and the roughness length for water vapour pressure and the roughness length for temperature (e.g. Braithwaite, 1995; King et al., 2008). This was the approach followed by Herrero et al. (2009) in the previous studies of the snow made in this study area and it is maintained in this research.



Regarding $K_{H0}$, a parameter also related to the turbulent heat fluxes, Tarboton and Luce (1996), Cline (1997) and Dingman (2002) ignore it in their formulas, while Jordan et al. (1999) and Koivusalo and Kokkonen (2002) stress its importance, and assign to it an even greater value than that predicted by the theory in accordance with the measurements obtained. In Herrero et al. (2009), this parameter turned out to be essential to simulate the processes correctly, and so it is consistently incorporated in the modelling.

In this study, the stability correction factors for non–adiabatic temperature gradients (in Eq. (A3) and Eq. (A4)) were not included, as their contribution to improving the accuracy of the results has proven inconclusive to date (e.g. Braithwaite, 1995; Tarboton and Luce, 1996; Hock, 2005; Herrero et al., 2009). Therefore, the model considers neutral buoyancy, at adiabatic lapse rate, which is consistent with the idea that on a mountainous hillside with significant slopes, stable atmosphere states do not develop in the way they do in valley areas. In the former, the boundary layer is more prone to mixing due to katabatic downhill flowing winds that are generated even under calm clear–sky conditions (Barry, 1992). According to Braithwaite (1995), uncertainty in $z_0$ may cause larger errors than neglecting stability. $K_{H0}$ and $z_0$ have been maintained as calibration parameters, this time estimated from direct measurements of evaposublimation and snowmelt instead of from snow depth and density values, as was done in Herrero et al. (2009).

## 3.2 Snow field surveys

Ten different daily field campaigns were carried out during the period 2009–2015 throughout Sierra Nevada, both on the south and north faces, to measure ablation from the snow, that is, the changes in the weight of a snowpack due to evaposublimation, condensation and melting. Each campaign lasted from 3 to 18 hours, and they were divided into 15 single meteorological states during which stationary or quasi–stationary meteorological conditions prevailed. They were conducted under different meteorological and topographical conditions, in different seasons, and with different types of snow, in order to achieve a representation of the states of the snowpack.

Evaposublimation (and condensation) and melting from the snowpack were measured using a modified version of the evaporation pan method with lysimeter. This method, also known as gravimetric, involves measuring the changes in mass of a finite volume of snow. Snow evaporation pans have been used for over forty years by, for example, Doty and Johnston (1969), Föhn (1973), Lemmelä and Kuusisto (1974), Bengtsson (1980). It must be noted that, under precipitation or windy conditions, which can blow the snow into or out of the pan, there may be a variation in the mass measured in the control volume that will not be directly distinguishable from evaposublimation unless it is measured separately.

The experimental device used in this study was developed following Avery et al. (1992) with some particular adjustments. It consists of two–tiered trays of white HDPE, the upper one being filled with undisturbed snow exposed to evaposublimation and condensation on its surface. The lower tray collects and protects from further evaporation the water that melts and percolates through the snow sample and through several holes drilled in the base of the upper pan in a regular grid. Both pans can be weighed jointly and separately, so evaposublimation or condensation and melting can be measured at the same time. This snow "ablameter" device has an exposed surface of 1260 (29.1×43.3) cm$^2$ and a depth of approximately 8 cm. Kaser (1982) and Valeo et al. (2005) used a similar device with only one pan, made of acrylic glass and aluminium respectively, both with small



surface areas of 400 and 260 cm$^2$ but they encountered some limitations due to this reduced size and the accumulation of meltwater in the pan. Froyland et al. (2010) also used just one transparent container made of acrylic glass with an exposure area of 700 cm$^2$, with which they could not measure the melting snow. The device used here is closer to the quasi–lysimeters used by Radionov et al. (1997) or Jackson and Prowse (2009), who placed two stacked trays to measure meltwater. Avery et al. (1992) also developed a similar design that they called "sublimimeter", made of insulation foam and Teflon–coated steel, able to hold a snow volume of $35 \times 35 \times 10$ cm$^3$.

The size of the trays selected for these experiments is as large as possible within the constraints of handling and weight. In this way, the snow contained is less affected by the small scale eddies in the wind field caused by the discontinuity in the surface of the snow, which may play a significant role in the sublimation rate (Earman et al., 2006). The change of mass and the weight of meltwater in the lysimeter were measured with a hand weighing scale Kern HDB (5K5) with a precision of 5 g, which for the 1260 cm$^2$ of the pan, gives us a resolution of 0.04 mm of SWE. During the snowmelt period, the setting–up and loading of the pans with snow is done early in the morning, when the snow structure is more stable. The snow probe is previously cut and then loaded into the pan by a sliding movement to keep the snow surface and structure as undisturbed as possible. In order to facilitate this operation, one of the short sides of the top tray can be removed. The pans are positioned with the top surface of the loaded snow flush with the original snow surface surrounding the pan, but safely separated from it to avoid mass exchange. In order not to disturb the snow properties, measurements are separated as much as possible in time (2–5 h). Measurement accuracy is preferred over its timing, keeping in mind that the understanding of the processes requires the same time resolution as the process itself, which in the case of evaporation may be hourly or higher (Lundberg, 1993). Each test lasts no more than 24 hours. On warm days with plenty of snowmelt, the test ends when the structure and surface of the snow samples begin to be unrepresentative of the surrounding snow.

Additionally, some snow properties required by the model were also measured during each field test. Albedo was measured using a hand pyranometer, Solar Power ISM400 (400–1100 nm $\pm5\%$), and snow density was estimated by gravimetric determinations of 1/3 litre core samples obtained in situ. Finally, the snow temperature was regularly measured during each test to establish the initial energy state of the snow and to check the correct performance of the snow model during the associated simulations.

### 3.3 Meteorological data during field surveys

During each snow ablation field test, extensive monitoring of the meteorological variables required by the model was performed by a complete weather station. The tests on the southern face of Sierra Nevada were carried out in the surroundings of the permanent weather station at the Refugio Poqueira monitoring site, at 2500 m.a.s.l. On the northern face, the field tests were located at different points throughout the area above the town of Pradollano, at altitudes ranging from 2400 to 2600 m.a.s.l., where a portable weather station was installed close to the ablameter device. In both cases, following the indications of Lundberg (1993), the ablameter was located upwind of the station to guarantee negligible disturbances of the meteorological conditions at the test point. Tabs. 1 and 2 show the main characteristics of the sensors installed in each weather station. At Refugio Poqueira, temperature and relative humidity were measured at 2.5 m above the snow surface, and wind speed at 3.0 m.





At the portable weather stations, these heights change to 0.6 and 0.8 m respectively. The air vapour pressure was determined by the standard psychrometric method. Finally, solar radiation was measured using a standard pyranometer in both cases, while downward longwave radiation could only be measured at the permanent station. Both stations were managed by Campbell Scientific dataloggers with a 1–sec frequency of measurement and 1 to 5–min averaging record of the outputs.

## 4 Results

The main purpose of the field surveys was two–fold: first, to measure the actual evapotranspiration rates in Sierra Nevada under different meteorological and snow conditions, and second, to provide meltwater and vapour rate data to validate the snow model performance. With the calibrated and validated model, a continuous point simulation at this site was performed to quantify the actual importance of evaposublimation in the snow ablation at different time scales, and its influence on the annual water balance.

### 4.1 Measurements of melt and evaposublimation

Table 3 summarises the main meteorological and snow mass fluxes measured for each of the 15 stationary periods identified during the field tests. The number of the field in the first column identifies the measurement test, 10 in total. Two of them (tests 8 and 10) are divided into 3 and 4 stationary periods each, because of the observed change of the meteorological forcing conditions (solar radiation, air or snow temperature, and even wind speed), mainly due to the transition from daytime to night time. The date and duration of each test is also recorded. The meteorological conditions over the test period are summarised as the average wind speed, relative humidity, temperature and shortwave radiation state (sunny "S", partially cloudy "C", overcast "O" or night "N"). The total melt and evaposublimation amounts measured with the ablameter are expressed in mm h$^{-1}$. Finally, information about albedo $\alpha$, and topography (slope, aspect, and face of Sierra Nevada, north or south) of the exact monitoring position is also shown.

At first, the evaposublimation measurement, though less intense, is more reliable than the melting measurement, as the latter relies on the correct drainage from the upper tray, that may sometimes be incomplete. We paid special attention to avoid the refreezing of meltwater in the drain holes, which was not observed in any of the performed tests. Three observations, one related to $M$ in test 9, and two other related to $E$ in tests 5 and 7, had to be rejected because they presented measurement errors.

The first thing that stands out is that, as expected, melt is a discontinuous phenomenon that occurs with increasing intensity as temperature rises, but mainly during daytime. Moreover, temperature was found to be a necessary but not sufficient driver for melting. As can be observed, there are some night tests with mean air temperatures of $5°C$ with no melt at all, since without the positive heat input from shortwave radiation, the sensible heat exchange cannot compensate for the cooling effect caused by longwave radiation and sublimation. Only one from a total of five night tests resulted in measuring melt (test 9), and the snowpack required a mean air temperature above $9°C$ for this to occur.



On the contrary, evaposublimation is a quasi–constant phenomenon, almost always found, albeit at low rates. Only in two of the tests (8a and 10b) was there a complete absence of measured vapour flux, that is, there was no change of weight in the trays. The first of them (test 8a) is a night time test over recent cold snow, at very low air temperature ($-9.3$°C) and with light wind ($2.4\ m\,s^{-1}$). These meteorological conditions are very similar to those encountered in the consecutive test 8b, which is

the only one with a measured net condensation (gain of mass in the trays) and with an intense rate ($0.036\ mm\,h^{-1}$ during 5.3 hours of test). Under the meteorological conditions in test 8a, carried out from 18:35 to 04:00, the expected behaviour is an initial sublimation while the snow cools from its "warmed" initial state to its night balanced state. Once the snow reaches its equilibrium temperature, the following condensation is the only mechanism to compensate longwave losses, which balanced in the end the initial loss of mass in the trays, as measured. The other test without noticeable $E$ is 10b, carried out on a sunny

day with high air temperature ($9$°C) and very low wind speed ($0.5\ m\,s^{-1}$), the lowest measured speed. These data may indicate that there is a threshold of wind that can inhibit the vapour fluxes; otherwise, these fluxes are always present. Even though their maximum rates were one order of magnitude under those for melting, accumulation of evaposublimation throughout the year may result in a significant total amount in the annual mass budget. Vapour loss rates above $0.04\ mm\,h^{-1}$ were measured equally on days under intense melting (test 2) and on cold wet days (test 3). Maximum evaposublimation rates of $0.11\ mm\,h^{-1}$

were measured under favourable weather conditions (cold days with low relative humidity).

### 4.2   Measurements vs. model estimates – test periods

Table 4 shows the results obtained when the energy balance snow model is applied over the test periods. The differences between measured and simulated melting/evaposublimation rates are also presented for each test as statistics for the error of the simulation. $E_{sim}$ only refers to the positive outgoing flux. When $E$ is simulated with a negative sign, it is taken as

condensation $C_{sim}$. The last column shows $z_0$, which was calibrated for each test in agreement with the actual conditions observed on the surface of the snow. The comparison of measured and simulated values of $E$, $M$, and the fraction of $E$ from the total ablation $E + M$ are plotted in Fig. 2. The goodness of the calibration is determined by using the mean error $ME$, the mean absolute error $MAE$ and the root mean square error $RMSE$. The agreement between model and observations for both $E$ and $M$ is very high, except for one outlier, which is circled. $RMSE$ for $E$ is $0.008\ mm\,h^{-1}$ ($6.7\%$ of the maximum $E$

measured), while for $M$ it is $0.009\ mm\,h^{-1}$ ($0.6\%$ of the maximum $M$). As for $E$ fraction, most values concentrate on 1 ($E$ without melting) perfectly matched by model simulations. Some inaccuracy is found in mixed states with simultaneous $E$ and $M$, with a final $RMSE$ of $0.013$ and a $ME$ of $0.04$ ($0.4\%$).

   The selected values for $z_0$ (Table 4) mostly range from 1.0 to 0.3 mm. These values are lower than the calibrated value in Herrero et al. (2009), 2.5 mm. Test 3 is the only one with a lower value, 0.1 mm, and it corresponds to conditions that promoted

the formation of a thin layer of ice on top of the snow, so its surface appeared particularly smooth. In general, $z_0$ under 0.5 mm is only measured on the north face of Sierra Nevada in tests performed in January or February. In this area at that time, snow is more likely to form surface ice layers with a noticeable influence on $z_0$. The average value of $z_0$ for all these field campaigns is 0.61 mm. The second calibration parameter of the model, $K_{H0}$, was found to perform correctly throughout all the cases if taken constant and equal to $1\ W\,m^{-2}\,K^{-1}$ which is in agreement with other studies (Jordan, 1991; Jordan et al., 1999), and





lower than the previously obtained value of $6\,W\,m^{-2}\,K^{-1}$ in Herrero et al. (2009). The inclusion of a decaying albedo instead of a constant value, together with the more accurate value of the longwave term in the energy balance equation, are some of the reasons behind these differences, together with the use of water fluxes as optimisation objectives instead of snow depth values. In the next section we test the model performance for snow depth under these calibrated values.

5      The five night time tests are particularly significant, as the absence of $K$ flux during the test period allows us to better adjust the calibration parameters. Also, the two measurement periods that extend throughout several continuous tests (8a–8b–8c and 10a–10b–10c–10d) show good agreement between measurements and simulations while maintaining the same calibrated parameters. This is a good indication of the validity of this calibration and the resilience of the model under different meteorological states.

10      Night test 8a, with 0 mm of measured $E$, is simulated by the model as a mixed state with an initial loss and a subsequent gain of vapour, reaching a balance by the end of the test. This is consistent with the hypothesis made in the previous section, even though the balance point is reached simulating a condensation rate much lower than that measured in the following hours, during test 8b. This test 8b stands out as an outlier because a high condensation rate (negative $E$) of -0.036 $mm\,h^{-1}$ is measured (circled in Fig. 2a) from 07:20 to 12:40, but it is simulated as only -0.004 $mm\,h^{-1}$ during the night interval. According to the 15 model, this condensation occurs from 04:00 to dawn, with a low air temperature ($-7.7°C$), not too high relative humidity (55%), and light wind (1.9 $m\,s^{-1}$), over very cold snow ($-17.5°C$). Once the sun rises, the model predicts sublimation as the snow temperature increases to balance with the air temperature. Unless proved otherwise, these results cannot be considered as a measurement error but a true deviation between measurements and modelling, even though the behaviour of the model on the whole seems consistent with that expected.

20      To estimate the sensitivity of the model to changes in $z_0$ within the ranges found in the field, we repeated the prior simulations considering a constant $z_0$ value equal to its average calibrated value. The results are plotted in Fig. 3. The error made when simulating with $z_0 = 0.61$ mm increases slightly in $E$ ($ME$ increases from 0.002 to 0.003 $mm\,h^{-1}$ and $RMSE$ from 0.008 to 0.012 $mm\,h^{-1}$). The increase in the errors in $M$ ($ME$ from 0.003 to 0.051 $mm\,h^{-1}$ and $RMSE$ from 0.009 to 0.164 $mm\,h^{-1}$) is concentrated in one particular test that moves away from the 1:1 line, though errors still remain low as $ME$ is only 25 3.4% of the maximum measured $M$. The instantaneous $E$ fraction from total ablation is barely affected by the change in $z_0$. It appears that the range of variation in measurements of $z_0$ is low enough not to affect the results significantly.

## 4.3 Annual simulation

On the basis of the above results, the energy balance model was used to simulate snow ablation processes at the Refugio Poqueira site from 2008/09 to 2014/15. Meteorological data from the Refugio Poqueira weather station were used to drive the 30 simulation. The parameters for this simulation were selected from the results of the validation achieved during the test periods. Thus, $K_{H0}$ and $z_0$ are supposed fixed and equal to 1 $W\,m^{-2}\,K^{-1}$ and 0.61 mm respectively.

     The validation of the snow model was assessed in terms of the snow depth observed at the Refugio Poqueira station. Fig. 4.a) shows how the model smoothly reproduced the main patterns in the intra– and inter–annual cycles in the snowpack. The correct simulation of accumulation and melting over time and the good match of the timing of the different intra–annual





melting cycles means that the model is computing adequately not only the snow depth and SWE time series, but also the cumulative energy balance of the snowpack (López-Moreno et al., 2013). Figs. 4.b) and 4.c) show the observed and simulated annual maximum accumulation and duration of the snowpack respectively. In both cases the simulated values fit well with the observations regardless of the wet/dry character of the year and the number of snowmelt cycles encountered. For the period

2009/10–2012/13, the maximum snowpack depths measured ranged from 627 to 1400 mm, while the number of cycles with total disappearance of the snow varied between 3 and 7. In general, autumn and spring precipitation events fall as snow but melt within a few days. However, there are drier or warmer years when complete melt is reached even during the winter. Snowpack duration ranged from 85 to 138 days. In the simulation, the Mean Error $ME$ and the Root Mean Square Error $RMSE$ were -81 and 235 mm for daily snow depth, 63 and 136 mm for annual maximum depth, and 2.8 and 10.5 days for annual snowpack

duration.

Fig. 5 shows the course of simulated cumulative snowfall, SWE, evaposublimation and snowmelt for each one of the seven hydrological years under study. The average annual ratio of total evaposublimation versus total ablation is 30.6%, oscillating from a minimum of 24.2% in 2010/11 to a maximum of 32.8% in 2014/15. The year with the highest total vapour loss is 2009/10, with 204 mm, while the year with the lowest loss is 2011/12 with 94 mm. Despite this low value, the percentage of

evaposublimated snowfall is high for this year, 29.4%. In general, years with low total snowfall result in a higher percentage of evaposublimation and a higher number of melting cycles. The model also estimates a mean annual condensation of 0.9% of the total snowfall (4.8 $mm\ yr^{-1}$).

Evaposublimation occurs steadily whenever there is accumulated snow, with less intensity towards the spring with the general rise in temperatures. Conversely, snowmelt shows a more intermittent behaviour, taking place only during the periods

of warmer weather during the winter and with dominating intensity during the spring. This persistence of vapour losses is illustrated in Fig. 6, where the probability density function (pdf) for 5–min values of $E$ and $M$ are plotted. The maximum rates for $M$ are approximately one order of magnitude higher than the rates for $E$, although $M$ is 0 or very close to 0 for almost 90% of the time with snow cover, compared to the 40% associated with zero values of $E$.

Fig. 7 represents the ratio of the evaposublimation versus total ablation in each month of the year, averaged over the entire

simulated period. The graph highlights the seasonal variations of the percentage of evaposublimation on an annual basis. In the months of December, January, and February, evaposublimation accounts on average for 47–51% of all the ablation that takes place. Snowmelt is present in these winter months, but with a moderate intensity (Fig. 8). During the following months, the evaposublimation ratio decreases sharply as snowmelt dominates, decreasing to approximately 22% in March, 12% in April and 4% in May. Monthly $M$ always dominates when compared to $E$ (Fig. 8), increasingly during the spring months. However,

the standard deviation of $M$ is always higher than that of $E$, and it has the same order of magnitude as the mean itself, which means that zero melt can be expected in every month but January, March and April. In contrast, monthly $E$ is less variable and shows lower standard deviations.

The cumulative annual energy fluxes in $W\,m^{-2}$ for the period 2008/09–2014/15, together with their mean and standard deviation proportion, are shown in Fig. 9. Warming fluxes $H$ and $K$ are on the positive side of the $x$ axis while cooling fluxes

$U_R$, $U_E$, and $L$ appear on the negative side. Their average fractions of the energy balance are 60% for turbulent sensible





exchange ($H$) and 40% for shortwave radiation ($K$) as positive (warming) fluxes; -54% for longwave radiation ($L$), -32% for turbulent latent exchange ($U_E$), and -14% for advective heat associated with precipitation ($U_P$) as negative (cooling) fluxes. The standard deviations are small, compared to those from the mass fluxes, $L$ being the most variable flux with 7%. Even though $L$ dominates on average over $U_E$ as a negative flux, there is one particular year (2010/11) in which both fluxes are

equal. The ratio between $H$ and $K$ also changes moderately between years.

## 5   Discussion

The turbulent heat transfer terms are probably the most uncertain contribution to solving the energy budget over the snow. The validity of the application of boundary layer theory to determine the turbulent fluxes over the snow, especially on complex mountainous terrains, is not clear (Hock, 2005). In general, snow in mountainous areas must always be considered as a non–

uniform surface, either because of the presence of patchy snow, obstacles such as rocks or shrubs that stand out of the snow surface on shallow snowpacks, or topographic gradients themselves. Besides the usual wind exposure of higher elevations, gravity winds usually develop even during calm days, promoting turbulent transfer under every meteorological condition (Feick et al., 2007). These turbulent terms include the calibration parameters used in the energy balance modelling, one referring to the sensible heat exchange ($K_{H0}$) and the other to both the sensible and latent heat exchange ($z_0$). The determination of both

parameters, together with the consideration of stability effects, are the major challenges of the physically–based snow models. These calibration parameters appear to be very site–dependent, according to the wide spectrum of variation described in the literature. In this work, these coefficients have been calibrated for Sierra Nevada (Spain) at approximately 2500 m.a.s.l. using detailed measurements of mass fluxes $E$ and $M$, with a final result of 1 $W\,m^{-2}\,K^{-1}$ for $K_{H0}$ and 0.61 mm for $z_0$ as a seasonal average value. Despite the large uncertainty that still exists regarding the roughness length of the different types of

snow surfaces (Martin and Lejeune, 1998), the measurements for this study area suggest a range of $z_0$ from 0.1 mm, for very smooth icy surfaces, to 1.0 mm on metamorphosed snow that shows surface forms as snow cups.

   The final value of the evaposublimation rate calculated from the snow surface is directly related to the latent heat flux, so the uncertainty associated with this turbulent phenomenon is carried forward to the estimation of $E$. The measurements confirm that the evaposublimation rate is small in magnitude (up to 0.11 $mm\,h^{-1}$) compared to snowmelt (up to 4.2 $mm\,h^{-1}$) but it

is very continuous over time and acts under virtually all weather conditions. Only in one of the 15 measured meteorological states did evaposublimation appear to be inhibited: on a very warm windless day, with major snow melting. The simulations performed with the snow model confirm the continuity of vapour loss throughout the year and between years, with a mean rate of 1.2 $mm\,d^{-1}$ (equivalent to 0.054 $mm\,h^{-1}$) and maximum rates of 7.2 $mm\,d^{-1}$ and 0.49 $mm\,h^{-1}$. As for melting, the simulated mean rate was 2.7 $mm\,d^{-1}$ (equivalent to 0.12 $mm\,h^{-1}$), while the maximum rates were 39.6 $mm\,d^{-1}$ and

4.8 $mm\,h^{-1}$. Maximum evaposublimation rates are reached during very windy periods, with maintained speed values above 8 $m\,s^{-1}$ and temperature close to 0°C. The relative humidity does not halt the process as long as it remains below 80 %, a value that indicates its supporting role to the wind effects.





Our evaposublimation rates are somewhat higher than those measured by Kaser (1982), who found maximum sublimation rates of 2 $mm\,d^{-1}$ in the Alpine summer at 3000 m.a.s.l., some way balanced with a correspondingly high condensation overnight. However, in a high latitude area like Canada, Valeo et al. (2005) recorded maximum values of sublimation equal to 6.3 mm in 8 hours (0.8 $mm\,h^{-1}$ on average) in Alberta (51°N), while Jackson and Prowse (2009) simulated mean vapour loss

of 0.4 $mm\,d^{-1}$ in open sites at Okanagan Basin (49°N) with SNTHERM (Jordan, 1991). The latter also simulated maximum melting rates of 40.5 $mm\,d^{-1}$, similar to the values found in this study. Even at this latitude, events of warm and dry air masses can occur during winter (the Chinook, popularly translated as "snow–eater", is an example of foehn winds), which occasionally enhance sublimation losses from the snow up to these values. In warmer areas, conditions to record high sublimation losses are easier to find. Avery et al. (1992) measured a maximum evaposublimation loss of 8.5 $mm\,d^{-1}$ under clear, dry and windy

conditions on the Colorado Plateau of Arizona at 35°N and 2100 m.a.s.l. In any case, modelled or simulated, it is reported than this rate is highly variable depending on the local conditions of the wind regime (Feick et al., 2007), on the meteorology and, therefore, on the time of the year when the snowpack accumulates.

During our field measurements, one of the tests showed a strong condensation rate of 0.036 $mm\,h^{-1}$ that could not be simulated with the model. The simulation of hoar growth in complex terrain is a difficult task since it demands high resolution

data of the wind regime, including thermally and topographically induced winds (Feick et al., 2007). Further measurements and study are necessary to establish whether this condensation rate is a common phenomenon in the area, where problems with hoar and ice over structures (for example, at ski resorts) are often reported.

Total annual evaposublimation is estimated as 24–33% of the total ablation of the snow, which represents a significant fraction. This result confirms the previous estimations made by Herrero et al. (2009), reached without direct measurement

of this water flux, and highlights the advantage of using physical models in approaching these processes. The difference in the annual evaposublimation between different climatic zones is related to the availability of those meteorological states that favour evaposublimation. In Sierra Nevada, evaposublimation is present almost continuously to a greater or lesser extent, so its contribution becomes important on an annual basis. The worst–case scenario for high evaposublimation rates takes place when the snow pack accumulates early in the season (Avery et al., 1992). The importance and variability of evaposublimation in Sierra

Nevada agrees with other studies in warm and semiarid mountainous regions around the world, such as California (20% (Marks and Dozier, 1992), 36% (Leydecker and Melack, 2000)), Colorado (15% (Hood et al., 1999), 17–43% (Froyland, 2013)), Canada (40% (Gray and Prowse, 1993)), the Andean Altiplano (30–90% (Vuille, 1996)), the Atlas mountains (44% (Schulz and de Jong, 2004), 7–20% (Boudhar et al., 2016)), and Israel (46–82% (Sade et al., 2014)), although its exact proportion depends greatly on the exact location of the sampling point and its altitude. At the Refugio Poqueira study site (2500 m.a.s.l.,

37°N), and in most of Sierra Nevada in general, year–to–year climate variability in precipitation and temperature interact non–linearly to allow the development of a highly heterogeneous snowpack, which leads to the corresponding variability in the percentage contribution of evaposublimation to total snow ablation.



## 6    Conclusions

In this study we have quantified the evaposublimation rates and the rest of the energy balance terms of the seasonal snowpack in the semiarid region of Sierra Nevada (37°N). The 15 field tests performed succeeded in validating the physically based snow model designed by Herrero et al. (2009). Although the measurement based on manually weighed trays is a traditional and not automated method, it achieves high accuracy thanks to the technical characteristics of the current weight measuring instruments. The measurements have confirmed the constant presence of evaposublimation from the snow in this semiarid environment, detecting maximum rates of 0.11 $mm\,h^{-1}$ in periods that were neither particularly dry nor particularly windy. Melting snow on warm days can reach much higher rates, up to 4.18 $mm\,d^{-1}$, but its effect takes place in shorter periods than those affected by evaposublimation, which is more persistent for different meteorological conditions. Throughout the period when the snow is accumulated on the soil surface, evaposublimation occurs during 60% of the time, while snowmelt only occurs during 10%. With these data, the energy–balance snow model was calibrated using two main parameters, both associated with turbulent fluxes: the aerodynamic roughness length of the snow $z_0$ and the sensible–heat transfer coefficient in windless conditions $K_{H0}$. $K_{H0}$ was set to 1 $W\,m^{-2}\,K^{-1}$, while $z_0$ was found to range between 0.1 mm over recent snow with an icy surface and 1.0 mm over mature snow with big grain size and an irregular surface. The mean value for $z_0$ was 0.61 mm.

The model satisfactorily reproduced the evaposublimation and melting rates during the monitored periods. Situations with simultaneous melting and evaporation were also correctly simulated. From these results, the continuous performance of the model at the Refugio Poqueira monitoring site (at 2500 m.a.s.l.) during the 2008-2015 period, produced an estimated evap-osublimation value between 94 and 204 mm per year, from a total snowfall of 320 to 676 mm per year, which accounts for between 24 and 33% of the total annual accumulated snow. On a daily basis, the evaposublimation rate reached a mean value of 1.2 $mm\,d^{-1}$; a maximum of 7.2 $mm\,d^{-1}$ with hourly peaks of 0.49 $mm\,h^{-1}$ being simulated on very windy days.

Regarding the energy balance, we were able to estimate that 32% of the cooling energy is due to the latent–heat transfer term, which is significant. Wind is the key element that establishes the final weight of this term in the energy budget for every season. Due to its proximity to the sea, and its high altitude compared to the neighbouring mountains, Sierra Nevada is a wind–exposed mountain range, which explains the relevant influence of this term on the snow regime over the mountain range.

The annual energy and mass balance over the snowpack is sensitive to small changes in variables governing the weather regime and/or their timing. Due to the elevational gradients and the seasonal and annual climate variability, high variability of the weather drivers can usually be found both spatially and over time in semiarid high mountain environments. Since simultaneous and intense monitoring of the snowpack is not feasible over these areas, the availability of a reliable snow model to infer the distribution of evaposublimation throughout Sierra Nevada, and to further simulate the evolution of the snowpack is an important and useful tool. In these regions, the impact of this return of water to the atmosphere is appreciable on the hydrology and on the availability of water as a resource. The results shown in this study are a first and essential step for estimating the influence of snow dynamics on runoff and water storage in the area, and for assessing water planning in the short and medium term. The implications for adaptation strategies are also relevant in a scenario of change in the energy budget regime.



## 7 Data availability

All the data used in this study can be provided by the corresponding author upon request.

## Appendix A: Snow mass and energy balance equations

The instantaneous mass and energy balance in the 1–layer control volume per unit of surface area is described as follows:

$$\frac{dm}{dt} = R - E + W - M \tag{A1}$$

$$\frac{d(m \cdot u)}{dt} = K + L + H + G + R \cdot u_R - E \cdot u_E + W \cdot u_W - M \cdot u_M \tag{A2}$$

where $m$ denotes the water mass in the snow column (snow water equivalent $SWE$), and $u$ is the internal energy per unit of mass ($U$ for total internal energy). Regarding mass fluxes, $R$ denotes the precipitation; $E$ represents water vapour diffusion (evaposublimation/condensation); $W$ is the mass transport due to wind; and $M$ is the melting water. The energy fluxes are the solar or shortwave radiation, $K$; the thermal or longwave radiation, $L$; the exchange of sensible heat with the atmosphere $H$; the heat exchange with the soil $G$; and the advective heat terms associated with each of the mass fluxes ($u_R$, $u_E$, $u_W$ and $u_M$) which refer to the unitary internal energy of each of the mass fluxes considered.

Eq. (A1) and Eq. (A2) form a coupled set of first order, nonlinear ordinary differential equations. They are solved with a first order finite difference approximation with a 5–min step time and an iterative algorithm (Herrero et al., 2009) that can reduce the time step in situations of numerical instability.

The turbulent energy diffusion terms for water vapour $U_E$ as well as for sensible heat $H$ can be resolved by basing calculations on the physics of turbulent transfer near the ground, as explained, for example, in Dingman (2002):

$$U_E \quad = E \cdot uE = \frac{K_{UE}}{\Phi_M \Phi_V} v_a (e_{sn} - e_a) \tag{A3}$$

$$H \quad = \left( \frac{K_H}{\Phi_M \Phi_H} v_a + K_{H0} \right) (T_a - T_{sn}) \tag{A4}$$

where $K_{UE}$ is the bulk latent–heat transfer coefficient; $K_H$ is the bulk sensible–heat transfer coefficient; $K_{H0}$ is the sensible–heat transfer coefficient in windless conditions ; $v_a$ is the wind speed at the reference altitude; $e_a$ is the air vapour pressure at the reference altitude; $e_{sn}$ is the saturation vapour pressure for the snow temperature, $T_{sn}$; $T_a$ is the air temperature at the reference altitude; and $\Phi_M$, $\Phi_V$ and $\Phi_H$ are the stability–correction factors for non–adiabatic temperature gradients, introduced to account for the buoyancy effects that may enhance or dampen the turbulent transfers because of the temperature gradient over the surface. There are numerous empirical expressions for these correction coefficients, depending on the Richardson number or the Monin–Obukhov length (Price and Dunne, 1976; Anderson, 1976; Oke, 1987; Cline, 1997; Jordan et al., 1999). However, application to actual wind and temperature conditions may lead to values of these coefficients that fall outside the limits for which they were empirically defined. This fact, as well as the uncertain improvement in the accuracy of the results



with their application (e.g. Hock, 2005), has led certain authors to reject them completely (Tarboton and Luce, 1996; Herrero et al., 2009), or limit their use to smaller ranges (Koivusalo and Kokkonen, 2002).

$K_{UE}$ and $K_H$ are defined as follows:

$$K_{UE} = u_E 0.622 \frac{\rho_a}{P_a} \frac{k^2}{\left[ln\left(\frac{z_a - z_d}{z_0}\right)\right]^2} \tag{A5}$$

$$K_H = \rho_a c_a \frac{k^2}{\left[ln\left(\frac{z_a - z_d}{z_0}\right)\right]^2} \tag{A6}$$

where $\rho_a$ is the mass density of air in $kg \cdot m^{-3}$; $c_a$ is the heat capacity of air (at constant pressure, $0.001005 \ MJ \cdot kg^{-1} \cdot K^{-1}$); $P_a$ is atmospheric pressure in kPa; 0.662 is the ratio between molecular weight of air and molecular weight of water vapour; $k$ is the dimensionless von Karman's constant (0.4); $z_a$ is the height at which wind velocity is measured, $z_d$ is the zero–plane displacement (0 for snow and ice) and $z_0$ is roughness height. $z_0 + z_d$ is the nominal surface level at which logarithmic boundary layer profile predicts zero velocity. As a consequence of the Ideal Gas Law, mass density of air decreases in altitude with atmospheric pressure $P_a$. Thus, from $P_a$ (kPa), $T_a$ (K), and the gas constant for air $R_a$ (0.288 for the units given), $\rho_a$ can be calculated as follows:

$$\rho_a = \frac{P_a}{T_a R_a} \tag{A7}$$

The $K_{UE}$ term includes the unitary internal energy $u_E$ advected to $E$, and it appears in Eq. (A2). If $e_{sn} > e_a$, evaporation occurs and $u_E$ is the advected heat of the water vapour that moves out from the surface layer into the air above measured. The internal energy of this flux as it moves out of the snow into the atmosphere will be that of water vapour with the temperature of the surface $T_{sn}$ with respect to the selected reference state. Therefore, the calculation of $u_E$ is indifferent to the initial state of water on the surface of the snow, and is the same for sublimation (with frozen surface) and evaporation.

$$u_E = \lambda_v + ce_v T_{sn}, \quad \text{if } e_{sn} > e_a \tag{A8a}$$

$$u_E = ce_v T_a, \quad \text{if } e_{sn} < e_a \tag{A8b}$$

where $\lambda_v$ is the latent heat of vaporization ($2.47 \ MJ \cdot kg^{-1}$) and $ce_v$ is the heat capacity of water vapour ($0.001850 \ MJ \cdot kg^{-1} \cdot K^{-1}$ at standard conditions $STP$). If $e_{sn} < e_a$ water vapour molecules enter the surface at $T_a$, where they condense. Their unitary internal energy will be that of Eq. (A8b).

The snow density $\rho_{sn}$ is mainly needed in the model for the calculation of the snow depth $h_{sn}$. It is considered uniform in the snowpack, and its evolution is calculated from an initial value for new snow $\rho_{sn-min}$ that is gradually modified in time through percolation, refreezing (both due to meltwater and rain), condensation, and new snowfall. There are two kinds of maximum density, one, $\rho_{sn-max}$, is due to grain growth associated with percolation, and the other, $\rho_{sn-frz}$, is a possible maximum density reached through internal refreezing of water. Density increase due to percolation is represented by a parametrisation that makes use of melting rate $M$, a growth coefficient $k_{\Delta\rho}$ with units of kg l$^{-1}$ mm$^{-1}$, and a normalized density $\Theta_{sn}$:



$$\Delta \rho_{sn} = M \cdot k_{\Delta \rho} \cdot \Theta_{sn}(\rho_{sn}) \tag{A9}$$

$$\Theta_{sn} = \frac{\rho_{sn} - \rho_{sn-min}}{\rho_{sn-frz} - \rho_{sn-min}} \tag{A10}$$

with $\rho_{sn-min} = 0.1$ kg l$^{-1}$, $\rho_{sn-max} = 0.5$ kg l$^{-1}$, $\rho_{sn-frz} = 0.65$ kg l$^{-1}$, and $k_{\Delta \rho} = 0.008$ kg l$^{-1}$ mm$^{-1}$ if $\rho_{sn} < \rho_{sn-max}$, 0 otherwise. These values were selected for agreement with the densities measured in the study area in Herrero (2007), which were between 0.05 kg l$^{-1}$ for new snow to 0.55 kg l$^{-1}$ for old melting snow, reaching up to 0.70 kg l$^{-1}$ when ice layers are present as a sign of major refreezing. After snowfall or condensation, $\rho_{sn}$ is recalculated as the weighted average of the old–snow density and $\rho_{sn-min}$.

*Acknowledgements.* We are grateful to Sergio Torres, Agustín Millares, Zacarías Gulliver, Marta Egüen, and Rafael Pimentel for their help in the field work. This study was funded by the Spanish Ministry of Economy and Competitiveness – MINECO (project CGL2011–25632, "Snow dynamics in Mediterranean regions and its modelling at different scales. Implications for water resource management" and project CGL2014–58508R, "Global monitoring system for snow areas in Mediterranean regions: trends analysis and implications for water resource management in Sierra Nevada").



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

**Tables**

| WXT510 Weather | Vaisala | Wind speed | 0?60 $\pm0.3\ m\,s^{-1}$ |
|---|---|---|---|
| Transmiter | | Air temperature | -52?+60 $\pm0.3°$C |
| | | Relative humidity | 0?100 $\pm3\%$ |
| | | Barometric pressure | 600?1100 $\pm0.5$ hPa |
| 109 Probe | CS | Snow Temperature (2 levels) | -50?+70 $\pm0.36°$C |
| CS300 | CS | Solar radiation | 300?1100 nm $\pm5\%$ |
| CR200 | CS | Datalogger | -40?+50°C |

**Table 1.** Sensors in the portable weather station used during the field surveys at the northern study sites and their technical characteristics.





| | | | |
|---|---|---|---|
| HMP45C | Vaisala | Air temperature | -40?+60 ±0.3°C |
| | | Relative humidity | 0.8?100 ±3% |
| RPT410F | Druck | Barometric pressure | 600?1100 ±0.5 hPa |
| SP–Lite | Kipp&Zonen | Solar radiation | 400?1100 nm |
| CGR3 | Kipp&Zonen | Longwave radiation | 4500?44000 nm |
| 05103–45 Alpine | Young | Wind vector | 0?60 m/s |
| CR10X | CS | Datalogger | -40?+50°C |
| T–200B w/Alter Shields | Geonor | Precipitation | 0–600 ±0.1 mm |

**Table 2.** Sensors in the permanent weather station at the Refugio Poqueira monitoring point and their technical characteristics.

| Test | Date | Duration (h) | Solar radiation | $\overline{W}$ ($m\,s^{-1}$) | $\overline{RH}$ (%) | $\overline{T}$ (°C) | $M_{obs}$ ($mm\,h^{-1}$) | $E_{obs}$ ($mm\,h^{-1}$) | $\alpha$ | Slope (°) | Aspect | SN face |
|---|---|---|---|---|---|---|---|---|---|---|---|---|
| 1 | 12–Mar–2015 | 8.6 | N | 3.2 | 32 | 4.0 | 0.00 | 0.041 | 0.53 | 15 | SW | S |
| 2 | 10–Apr–2014 | 4.6 | S | 2.1 | 34 | 10.5 | 4.18 | 0.044 | 0.62 | 23 | SW | N |
| 3 | 27–Feb–2014 | 2.4 | O | 7.2 | 81 | -1.0 | 0.00 | 0.041 | 0.8 | 3 | W | N |
| 4 | 28–Jan–2014 | 5.2 | S | 4.3 | 46 | -1.0 | 0.00 | 0.113 | 0.59 | 7 | NW | N |
| 5 | 23–Jan–2014 | 4.3 | S | 8.4 | 70 | -4.4 | 0.00 | – | 0.63 | 2 | W | N |
| 6 | 28–Nov–2013 | 3.3 | C | 1.3 | 86 | -3.4 | 0.00 | 0.020 | 0.75 | 20 | N | N |
| 7 | 15–Mar–2013 | 5.0 | S | 1.3 | 49 | 3.2 | 1.51 | – | 0.62 | 5 | S | S |
| 8a | 1–Mar–2011 | 12.8 | N | 2.4 | 62 | -9.3 | 0.00 | 0.000 | 0.8 | 8 | S | S |
| 8b | 2–Mar–2011 | 5.3 | S | 1.9 | 55 | -7.7 | 0.00 | **-0.036** | 0.8 | 8 | S | S |
| 8c | 2–Mar–2011 | 2.6 | S | 1.6 | 63 | -4.2 | 0.00 | 0.031 | 0.8 | 8 | S | S |
| 9 | 29–Apr–2010 | 10.1 | N | 3.6 | 47 | 9.3 | – | 0.015 | 0.45 | 2 | S | S |
| 10a | 10–Mar–2009 | 11.8 | N | 5.6 | 15 | 3.9 | 0.00 | 0.111 | 0.67 | 12 | S | S |
| 10b | 11–Mar–2009 | 3.5 | S | 0.5 | 46 | 9.2 | 1.19 | 0.000 | 0.67 | 12 | S | S |
| 10c | 11–Mar–2009 | 13.8 | N | 2.6 | 37 | 4.9 | 0.00 | 0.047 | 0.67 | 12 | S | S |
| 10d | 12–Mar–2009 | 3.0 | S | 2.4 | 25 | 5.0 | 0.11 | 0.025 | 0.67 | 12 | S | S |

**Table 3.** Summary of the different test periods with their date, duration, solar radiation state (N, night; S, sunny; O, overcast; C, cloudy), main weather drivers (W, wind speed; RH, relative humidity; T, air temperature), measured evaposublimation ($E_{obs}$) and melting ($M_{obs}$) rates, measured snow albedo ($\alpha$) and main topographic features of the test sites.





| Test | $M_{sim}$ $(mm\,h^{-1})$ | $E_{sim}$ $(mm\,h^{-1})$ | $C_{sim}$ (-$E_{sim}$) $(mm\,h^{-1})$ | Error $M$ $(mm\,h^{-1})$ | Error $E$ $(mm\,h^{-1})$ | $K_{sim}$ $(MJ\,h^{-1})$ | $L_{sim}$ $(MJ\,h^{-1})$ | $H_{sim}$ $(MJ\,h^{-1})$ | $U_{E\,sim}$ $(MJ\,h^{-1})$ | $z_0$ (mm) |
|------|------|------|------|------|------|------|------|------|------|------|
| 1 | 0.00 | 0.041 | – | 0.00 | 0.000 | 0.00 | -0.27 | 0.39 | -0.10 | 0.9 |
| 2 | 4.17 | 0.046 | – | -0.01 | 0.002 | 1.37 | -0.22 | 0.37 | -0.11 | 0.5 |
| 3 | 0.00 | 0.049 | – | 0.00 | 0.008 | 0.26 | -0.16 | 0.04 | -0.12 | 0.1 |
| 4 | 0.00 | 0.112 | – | 0.00 | 0.002 | 0.60 | -0.33 | 0.11 | -0.28 | 0.3 |
| 5 | 0.00 | 0.131 | – | 0.00 | – | 0.71 | -0.35 | -0.04 | -0.32 | 0.3 |
| 6 | 0.00 | 0.010 | 0.000 | 0.00 | -0.011 | 0.08 | -0.04 | 0.06 | -0.02 | 1.0 |
| 7 | 1.52 | 0.042 | – | 0.01 | – | 1.14 | -0.60 | 0.08 | -0.10 | 0.8 |
| 8a | 0.00 | 0.002 | 0.002 | 0.00 | 0.000 | 0.00 | -0.27 | 0.19 | 0.00 | 0.5 |
| 8b | 0.00 | 0.004 | 0.003 | 0.00 | 0.037 | 0.37 | -0.36 | 0.14 | -0.01 | 0.5 |
| 8c | 0.00 | 0.029 | – | 0.00 | -0.002 | 0.57 | -0.32 | -0.04 | -0.07 | 0.5 |
| 9 | 0.75 | 0.019 | 0.002 | – | 0.000 | 0.01 | -0.14 | 0.43 | -0.05 | 1.0 |
| 10a | 0.00 | 0.130 | – | 0.00 | 0.019 | 0.02 | -0.32 | 0.62 | -0.32 | 0.7 |
| 10b | 1.20 | 0.003 | – | 0.01 | 0.003 | 0.82 | -0.50 | 0.08 | -0.01 | 0.7 |
| 10c | 0.00 | 0.041 | – | 0.00 | -0.006 | 0.01 | -0.25 | 0.30 | -0.10 | 0.7 |
| 10d | 0.14 | 0.042 | – | 0.03 | 0.017 | 0.46 | -0.34 | 0.26 | -0.10 | 0.7 |

**Table 4.** Summary of the simulated results ($M$, melting; $E$, evaposublimation; $C$, condensation) from the energy balance model for each test period, together with the simulated energy flux terms ($K$, shortwave radiation; $L$, longwave radiation; $H$, sensible–heat exchange; $U_E$, latent–heat exchange) and the calibrated values for the model parameters ($z_0$, aerodynamic roughness length; and constant $K_{H0}= 1$ $W\,m^{-2}\,K^{-1}$, sensible–heat transfer coefficient in windless conditions).



**Figures**

**Figure 1.** Location of Sierra Nevada in southern Spain (left) and Digital Elevation Model (m) of the study area (right). The enlargement shows the limits of the Sierra Nevada National Park (white line) and the location of the Refugio Poqueira monitoring site (2500 m.a.s.l.) and the town of Pradollano (2100–2300 m.a.s.l.).

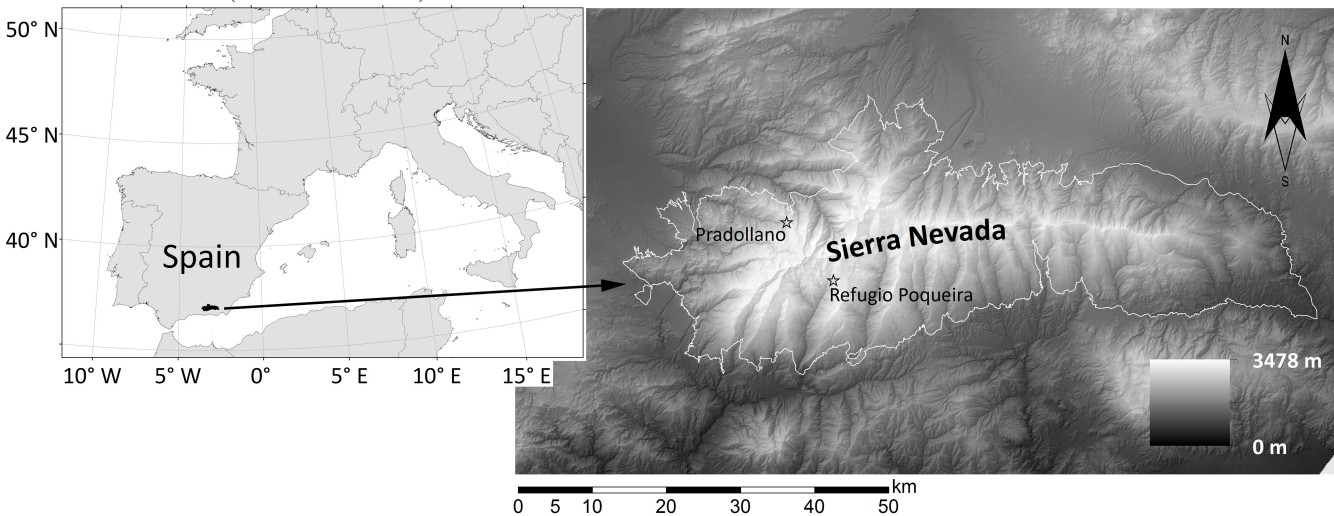

**Figure 2.** Measurements versus model estimates of evaposublimation ($E$), melting ($M$) and evaposublimation fraction from total ablation ($E/(E+M)$) for the different test periods using the calibrated values for each test in Table 4. The line indicates a 1:1 relationship between observations and simulated values.

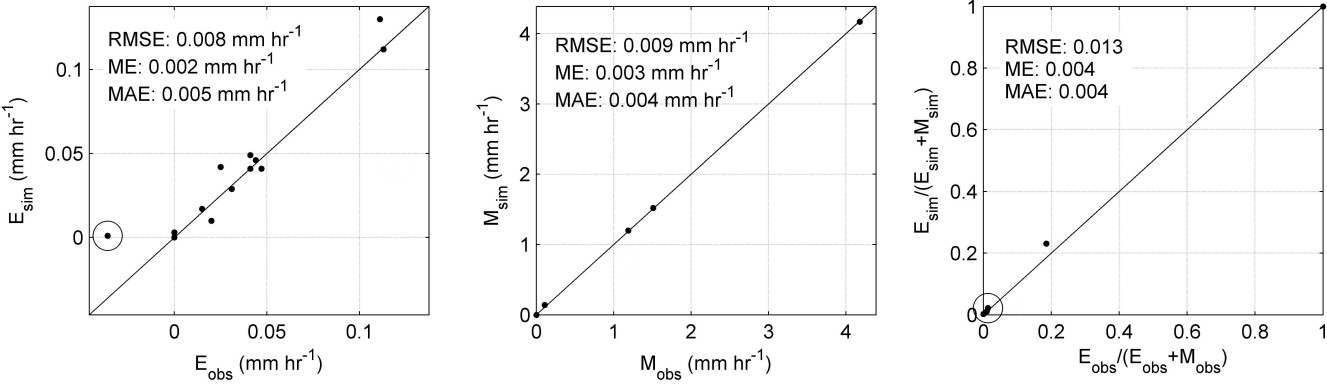





**Figure 3.** Measurements versus model estimates of evaposublimation ($E$), melting ($M$) and evaposublimation fraction from total ablation ($E/(E+M)$) for the different test periods using a constant $z_0$ of $0.61mm$. The line indicates a 1:1 relationship between observations and simulated values.

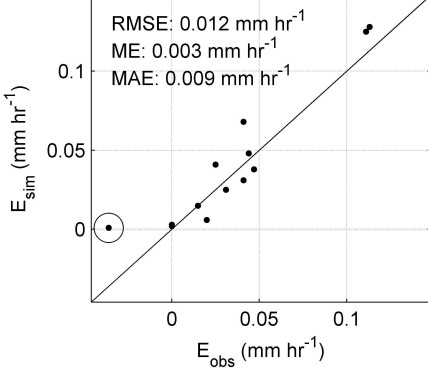
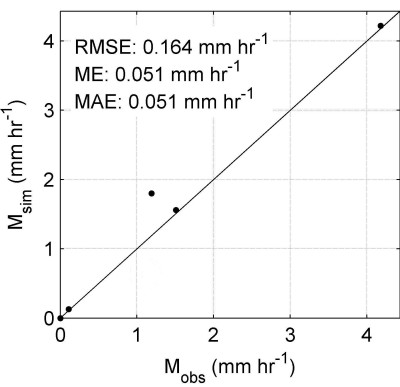
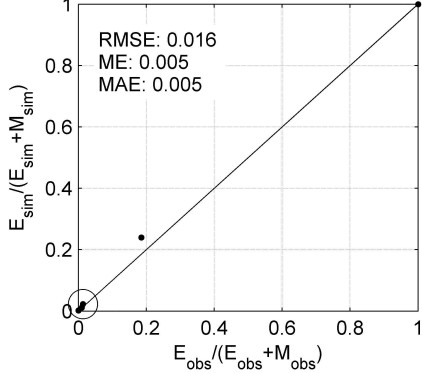



**Figure 4.** a) Snow depth in mm simulated for each snow season at the Refugio Poqueira site from the hydrological year 2008/09 to 2014/15. The grey crosses show the observed snow depth for years 2009/10 to 2012/13. b) Observed versus simulated annual maximum snow depth in mm at this site. c) Observed versus simulated annual duration of the snow depth in days at this site. The line at b) and c) indicates a 1:1 relationship between observed and simulated values.




**Figure 5.** Cumulative snowfall together with the stacked cumulative snowmelt ($M$) and cumulative evaposublimation ($E$) (in mm) for each snow season at the Refugio Poqueira site from the hydrological year 2008/09 to 2014/15. The white area between the snowfall and the stacked $M$ and $E$ represents the instant SWE during the year. The percentage at the end of every season indicates the ratio of annual evaposublimation compared to total ablation.

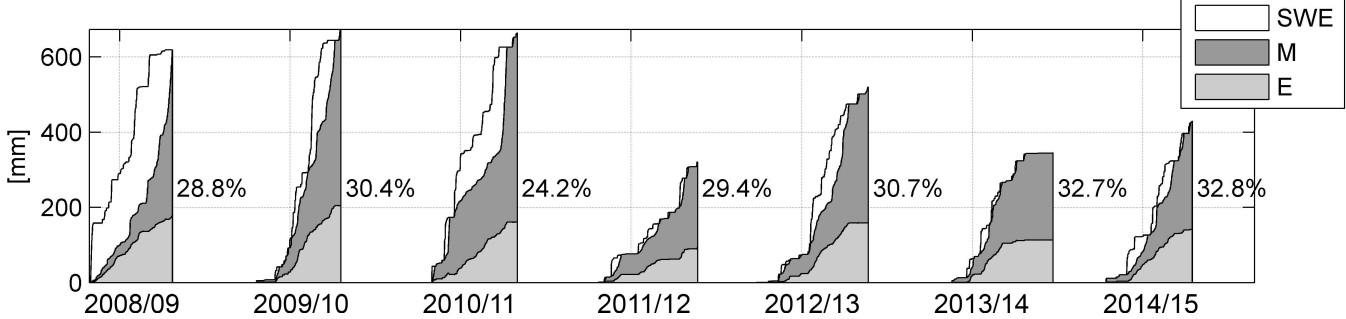

**Figure 6.** Pdf of the mean snowmelt ($M$) and evaposublimation ($E$) rates in mm 5–min$^{-1}$ at the Refugio Poqueira site from the hydrological year 2008/09 to 2014/15. The zoom on each plot shows the distribution without the influence of the zero–values.

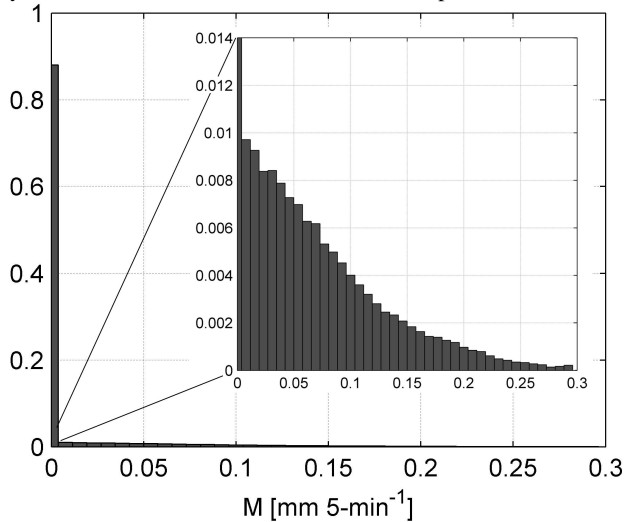
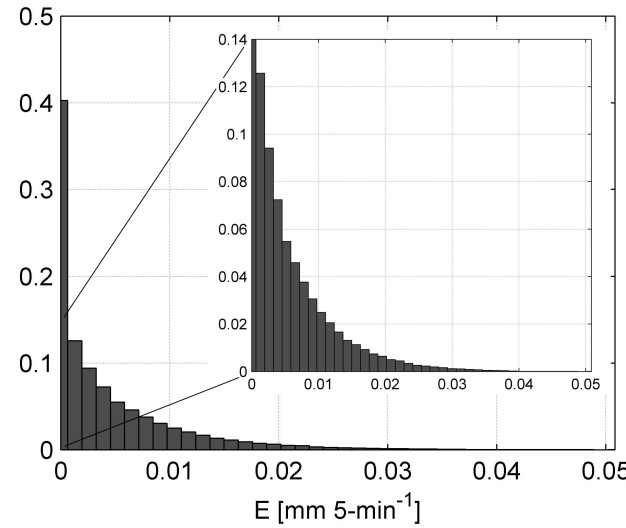



**Figure 7.** Mean monthly ratio of evaposublimation compared to total ablation during the year at the Refugio Poqueira site for the study period 2008–2015. Whiskers represent standard deviation from the mean.

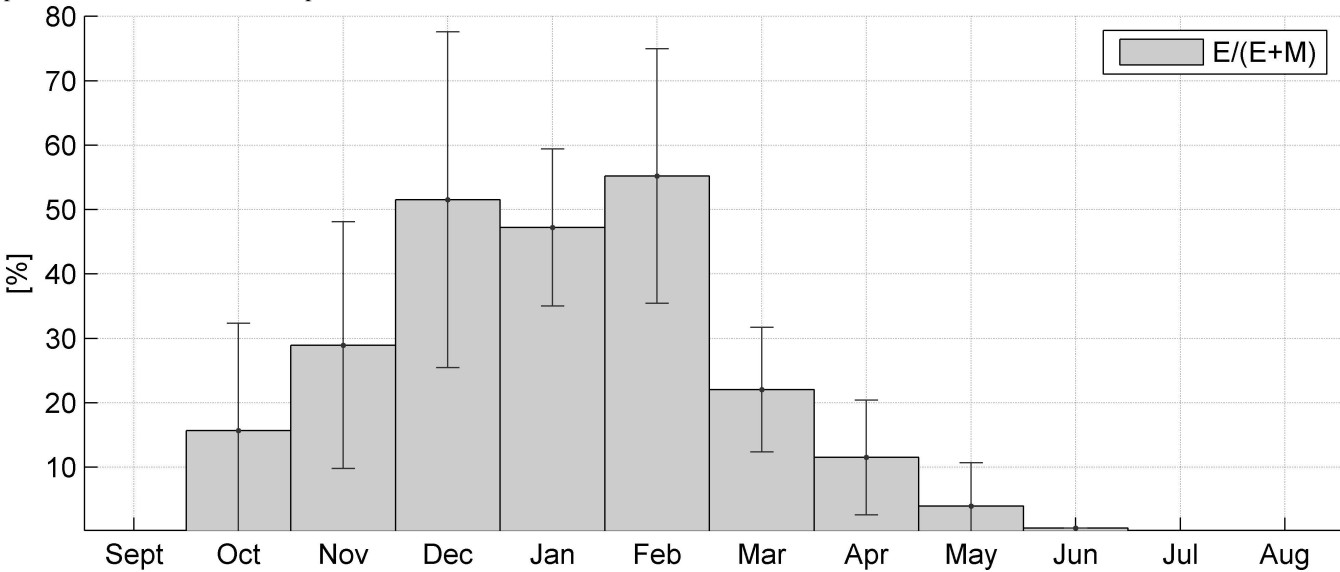

**Figure 8.** Mean monthly snowmelt ($M$) and evaposublimation ($E$) rates in mm month$^{-1}$ during the year at Refugio Poqueira site for the study period 2008–2015. Whiskers represent standard deviation from the mean.

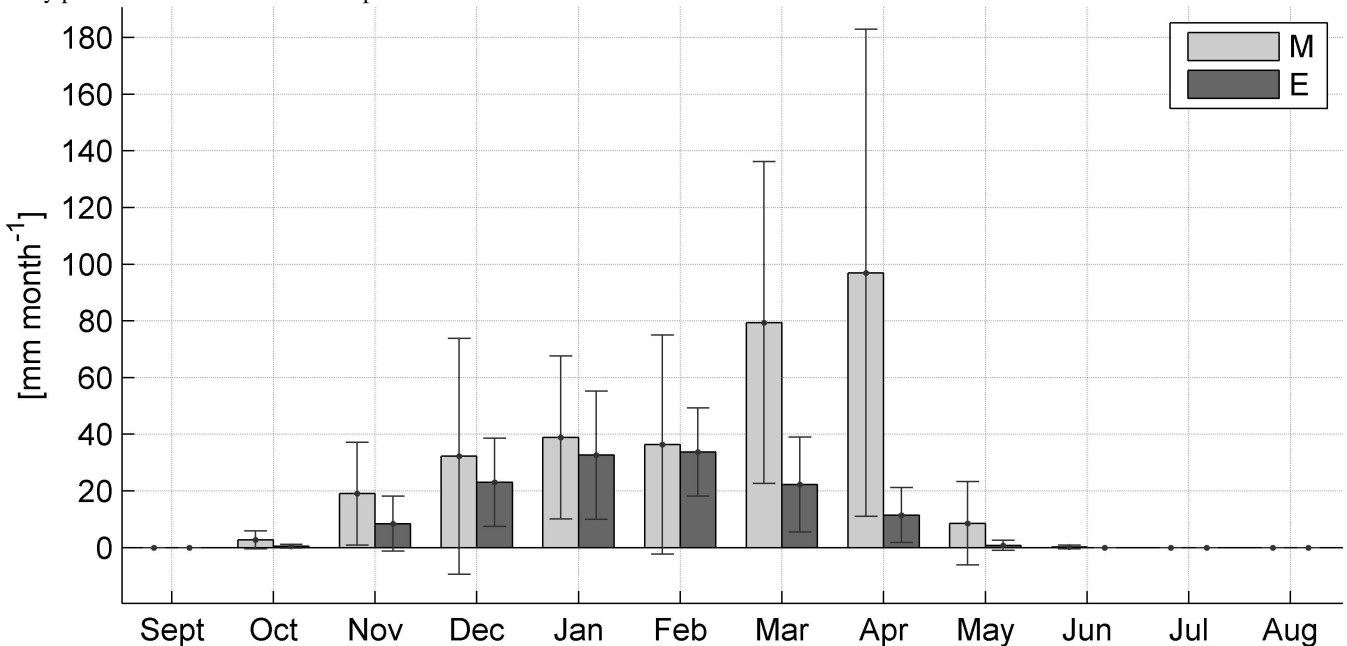





**Figure 9.** Annual mean energy balance ($W\,m^{-2}$) over the snowpack at the Refugio Poqueira site from the hydrological year 2008/09 to 2014/15 and average proportions of the fluxes, warming as positive and cooling as negative.

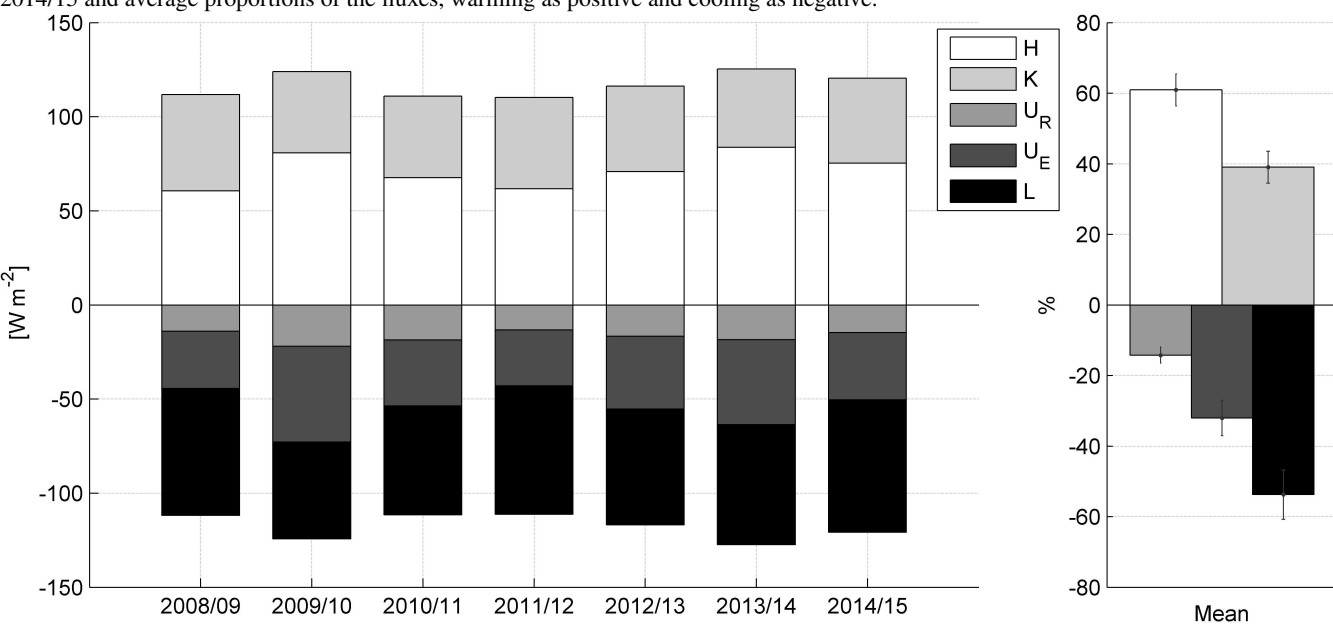





**Figure 10.** Mean monthly energy balance over the snowpack in $W\,m^{-2}$ during the year at the Refugio Poqueira site for the study period 2008–2015. Whiskers represent standard deviation from the mean.

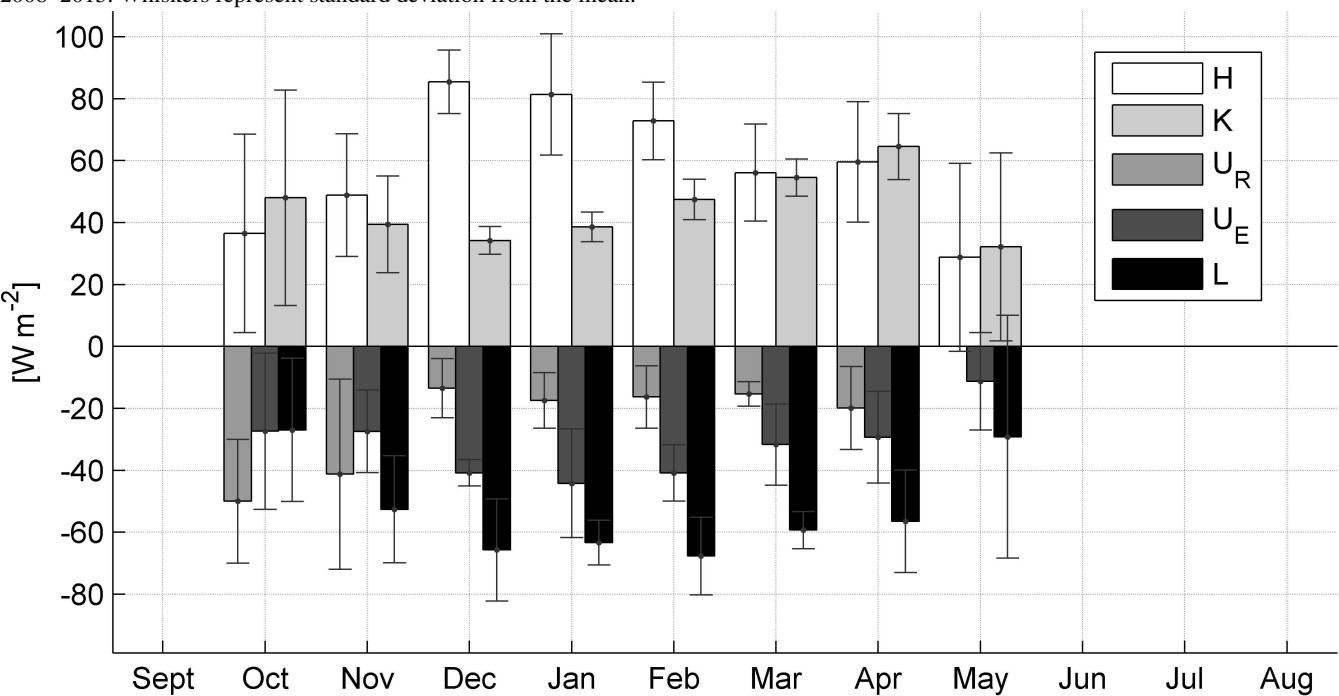