# Peer review of "Evaposublimation from the snow in the Mediterranean mountains of Sierra Nevada (Spain)"

_The Cryosphere, 2016_

## Referee Comment (RC1) · Anonymous Referee #1 · 23 Sep 2016

General comments:

This work details on evaposublimation rates from snow in the Spanish Sierra Nevada. It is based on direct observations over several years as well as numerical simulations using a snow model that has been trained to match observations. The study highlights the hydrological relevance of evaposublimation in meteorological conditions prevalent in the study area, in particular considering its frequency of occurrence.

Although there are quite a few studies that have reported on evaposublimation rates from other parts of the world, I do appreciate the effort put into these measurements. Its combined evaluation in conjunction with the model simulations is generally solid. Calibrating the snow model using fluxes rather than states (but using the latter for

validation) makes a welcome component. A missed opportunity is that there appears to be no systematic data available on snow surface characteristics that could have been compared to zo values presented in Table 4. Nevertheless this study should in my opinion be published after addressing the specific comments listed below.

Specific comments (reference is given to page / line numbers):

(2/26) Shouldn't "latter" be "former", if you include sublimation losses from snow intercepted in forest canopy?

(3/11) Why should this be a problem of the device? A snow lysimeter is to measure snowpack runoff, not snowmelt rates.

(3/25) You highlight the simplicity and low costs for traditional manual measurements versus the need for constant maintenance of automatic devices such as snow pillows. But isn't it the manual measurements that require constant maintenance?

(5/14) Please mention if any of your data were affected by snow-vegetation interactions.

(5/15) Section 3.1 is a bit lengthy to my taste, given that this is a published model; irrespective of modifications that should of course be described here or in the appendix.

(5/31) Will probably be handled by the type editor, but I suggest to refer to "Appendix A".

(6/16) You may want to add here that L_down is available for simulations at Poqueira, but not for Pradollano.

(6/35) But zo is not constant for simulations presented in Table 4.

(7/16) In the abstract you mention 15 field campaigns, here it is 10. Find a consistent wording to discriminate between campaigns (10) and data sets (15).

(7/22) You use the term evaposublimation referring to evaporation and/or sublimation, but for the revers process only the term condensation is used without specifically addressing possible deposition (resublimation). I suggest using consistent terminology throughout the manuscript (see also at 14/2)

(7/30) Consider adding a photo of your device. Why/how would the lower tray inhibit further evaporation?

(8/10) Accuracy is more relevant than precision.

(8/28) "Surroundings" is a bit unspecific, within how many meters of the station?

(9/1) Did you account for different instrument heights when modelling N/S sites?

(9/21) It seems contradictory to name E measurements more reliable if you have to omit 2 of those values over 1 omitted M value.

(9/24) Better to provide specific reasoning to delete those values from your results. Outlier removal is a sensitive matter.

(10/1) Reword "quasi-constant", this process is not quasi-constant, it may occur most of the time. Moreover, "60% of the time" doesn't seem to match "almost always".

(10/15) "favorable" for what, for evaposublimation? Then you should probably mention wind.

(10/20) Delete "in agreement with the actual conditions" or provide data, in particular if you have some!

(10/34) "correctly"? Moreover, did you allow $K\_H0$ to vary between experiments or did you force $K\_H0$ to be constant?

(11/8) The three highest errors in E stem from these 2 two periods, so I do not necessarily agree with this statement.

(12/1) The order of statements seems strange. Given that you used flux data to calibrate your model, it should primarily replicate the flux terms, and eventually also the states, not the other way around.

(12/34) Just out of curiosity, is G too small / irrelevant to be shown?

(13/25) Please restrict this statement to your field site or the meteorological conditions in the Sierra Nevada. Moreover, from Table 3 it seems there are 3 instances of observed zero or negative E.

(14/2) What/where is "in the Alpine summer"?

(14/14) What do you mean by "high resolution"? I don't agree with your reasoning. You observed condensation at Poqueira, this is where you do have local meteorological data. Speaking of surface hoar formation and associated mass fluxes: you may want to look at a Stössel et al (2010, doi:10.1029/2009WR008198).

(22/Table 1) This table is incomplete, please add a header and remove misprinted characters such as the "?".

(23/Table 2) This table is incomplete, please add a header and remove misprinted characters such as the "?".

(23/Table 3) "Solar radiation" seems not the best term for what is presented in the column below.

(25/Figure 2) The third panel on the right could be removed.

(26/Figure 3) The third panel on the right could be removed.

(27/Figure 4) Remove 2008/09 data, which is not a complete winter season, also considering that manual measurements commenced in 2009/10.

(28/Figure 5) Combining Figure 5) with Figure 4a) seems to indicate very low snow densities in years such as 2011/12 and 2013/14. Are these values correct?

(29/Figures 7-10) These Figures take quite some space, but there is comparably little text in the body of the paper associated with these figure. Consider deleting two of the figures or extending the associated text.

---

## Referee Comment (RC2) · W. Eugster (Referee) · 1 Oct 2016

The authors modeled snowmelt and evaporation/sublimation losses from a snow pack in the Sierra Nevada, Spain, using a point energy budget model over 7 winter seasons (2008/9 to 2014/15). Their model suggests that 24 to 33% of the annual ablation of the snow pack is not via meltwater, but via gaseous vapor losses ("evaposublimation").

Although I have some critical remarks, I find the paper sound, relevant and suitable for the journal after the necessary revisions.

My own background is rather in eddy covariance flux measurements which are not employed here, and hence some critical remarks relate to the fact that from reading

the paper I got the impression that the authors would have really profited from eddy covariance flux measurements, which are no longer as difficult to perform as e.g. Hock (2005) thought more than 10 years ago. There are probably 3 sites in Spain that might have data for follow-up studies:

Castellar de N'Hug, Spain, Pyrenees: https://fluxnet.ornl.gov/site/4055
Lanjaron, Spain, Sierra Nevada: https://fluxnet.ornl.gov/site/4060
Laguna Seca, Spain, Sierra Nevada: https://fluxnet.ornl.gov/site/4058

With eddy covariance one could directly measure sensible heat flux and thus the somewhat weakly justified assumption made by the authors that the turbulent transfer coefficient for sensible heat, $K_{H0}$ = 1 W m$^{-2}$ K$^{-1}$ could have been omitted. Moreover, $z_0$ could have been derived from the direct measurement of momentum flux and horizontal wind speed, and the bascially tuned value for $z_0$ of 0.61 mm (both given on page 11, line 31) would have led to a more vigorous testing of the model.

If you add latent heat flux measurements, then of course eddy covariance flux measurements become more demanding, but already a simple sonic anemometer would provide the information mentioned above.

Having said so, I still think that the authors did a good job with the approach they used and I hope that addressing the following major points helps to improve the paper before final acceptance.

**Major Points**

1.

2/21-22: "The evaposublimation rate depends on the vapour pressure gradient between the surface of the snow and the air, which is mainly influenced by the local wind

intensity, and hence, by the complex turbulent phenomena occurring in the boundary layer." – I think you should more clearly phrase that in the first place you need a lot of energy to evoporate or sublimate water. It is not primarily the vapor pressure gradient that drives the flux, it is the heat supply to the snow surface (which is of course related to all gradients). Please rephrase. Actually on line 30 on same page you have a remark about solar radiation, but not a general picture of the relevance of energy fluxes.

2.

3/21-22: You mention that eddy covariance flux measurements "are complex, fragile, and require large, clear, low–angle areas to function optimally". This is not really correct. The measurements actually function quite nicely, but the key issue is that they are point measurements, and although they might be very accurate point measurements, the relation of this point measurements to the surface area is a challenge. See point 3 below. As an example: why do large eddy simulation (LES) people simply use arrays of ultrasonic anemometers to obtain field data? Because they can use exactly such point measurements to validate their models.

3.

13/8-10: The authors write "The validity of the application of boundary layer theory to determine the turbulent fluxes over the snow, especially on complex mountainous terrains, is not clear (Hock, 2005)" – which sounds quite special. I double-checked the Hock (2005) reference and thus do not think that this wording can withstand a careful check. First, several statements in Hock (2005) are by now outdate, e.g. the idea that the eddy covariance technique "require[s] sophisticated instrumentation with continuous maintenance, which render them unsuitable for operational purposes. Consequently, such studies are rare and restricted to short periods of time". In the meantime

there are many such continuous measurements. Second, this means that all other statements are focusing on the flux-gradient method that she is interested in, thus it cannot be deduced that her judgement applies to all possible approaches. Third, the fact that some authors "lack an explanation in terms of boundary layer theory" (page 378, left column) does not mean that such a theory is not valid.

What would be an acceptable summarization of the Hock (2005) paper in this context could be written e.g. with the wording "Flux-profile and bulk transfer approaches have been shown to be problematic over sloping terrain to determine turbulent fluxes (Hock, 2005)". It is essential to make clear that it is not a problem of the theory, but of the flux-gradient or bulk method that Hock (2005) talks about. In principle one could use eddy covariance, but also there could be issues since this is a point measurement and the relation to the footprint area influencing that measurements is challenging (if you need a more detailed explanation then please consult Eugster and Merbold, 2015).

You use exactly that boundary-layer theory in your modeling approach (e.g. equations A3 and A4) and there you found a more appropriate wording for summarizing the information given by Hock (2005).

4.

Data availability: it would be great if the data could be placed in a long-term archive, such as www.pangaea.de (which is free of charge for the authors)

**Details**

1/9 and many more places: "m.a.s.l." has one period too much: there is never a period after m for meters. Thus "m a.s.l."

1/10: "The ratio is changeable" – do you mean "variable"? or what does this actually mean?

1/11: "timing of the meteorological inputs, generally unforeseeable in this semiarid region" – there is something wrong here. I do not really get what you want to express. The timing of INPUT sounds incorrect in this context, and thus I do also not understand the connection with "unforeseeable" (maybe you mean that forcasting weather conditions does not work in this semiarid region? But are you sure there is no skill at all in such forecasts?) - please rephrase.

1/16: "as the latitude descends" – please rephrase, the latitudes stay in place. My understand is that you wanted to say "at increasing altitudes with decreasing latidude".

2/13: only use the word "significant" in the contects of statistical significance tests. If it does not relate to statistics, then use other words that do not have a special meaning in scientific texts. But here there is an error: "significant data series" does not sound correct anyway. Maybe you wanted to say something about data availability (no gaps, long time series?)

2/15: "source of distributed data" – probably "source of spatial data"?

2/20: what do you mean with "latent heat balance"? Probably "latent heat FLUX"?

5/3: what is an "alter-shielded rain gauge"? Please explain in more detail or give a reference where I could inform myself about this term that I do not know.

5/20 and many places elsewhere: you seem to have had some trouble with the characterset and all these question marks most likely should have a specific meaning. Please search for all question marks in the text and make sure that in the revisions you get the correct characters everywhere.

6/4: do not use computer code writing in text passages. Here you should use $\leq$

6/19: add "Appendix" before A

6/25: you cite Calanca (2001), but he does not primarily look at the aerodynamic roughness length, but at the roughness length for temperature. This topic again is related to the issue that you did NOT use eddy covariance flux measurements. With EC flux measurements you would bypass this issue. The information that the Calanca (2001) reference relates to actually would rather fit the information on line 34, same page. Best would be to rewrite and make sure the confusion between aerodynamic roughness ($z_0$) and roughness length for temperature ($\theta_0$) is resolved.

7/11-12: "According to Braithwaite (1995), uncertainty in z0 may cause larger errors than neglecting stability." – this actually is a strong argument why you should try out eddy covariance in follow-up research! With EC you measure both $z_0$ and $z/L$ (or better: you can compute these two from the raw measurements).

8/22 and Tables 1 and 2: here is an error: the $\pm 5\%$ uncertainty does not relate to the range of wavelengths that the sensor is sensitive to, but to the units of measurements, which are W m$^{-2}$. All other sensors except radiation sensors in the tables have range of measurement in correct units $\pm$ uncertainty, please give the same for radiation sensors and specify their wavelength sensitivity elsewhere (e.g. for snow temperature you mention 2 levels, you could do the same for solar radiation and write 300–1100 nm in parentheses).

9/2: "The air vapour pressure was determined by the standard psychrometric method." – I am not convinced about this: the standard psychrometric method uses a dry bulb and wet bulb temperature sensor. You however do not mention a wet bulb sensor, but a relative humidity sensor on page 8, last line. Thus, you calculated vapour pressure differently - please correctly inform us how you calculated it from temperature and relative humidity (most likely you used some equation like the Magnus equation to determine saturation vapor pressure at air temperature, then used relative humidity to calculate actual vapor pressure).

9/2: "using a standard pyranometer in both cases" – I do not agree. In Tables 1 and 2 you show that you use a CS300 at one site and an SP-Lite at the other. Both are silicon photovoltaic detector sensors, that are calibrated against a standard pyranometer, but they are NOT standard pyranometers! Please reword. For more information: https://www.campbellsci.com/cs300-pyranometer, http://www.kippzonen.com/Product/9/SP-Lite2-Pyranometer

9/27: "Moreover, temperature was found to be a necessary but not sufficient driver for melting." – In fact, it is the sensible heat flux, which is a function of the temperature GRADIENT. Please be more precise in your wording.

9/28 and elsewhere: you are not consistent in how you print physical units such as °C or $\mathrm{mm\,h^{-1}}$, sometimes in italics, sometimes not. Please homogenize (see the guidelines)

9/29: "positive heat input from shortwave radiation" – this is another shortcut that students tend to misunderstand. Please reword and make sure it is clear that shortwave radiation is a high-level form of energy which first needs to dissipate to heat, but shortwaver radiation by itself is NOT a heat input.

10/1: what is meant with "quasi-constant"? is it "continuous" (in opposition to sporadic)?

10/20: how did this calibration go for $z_0$? This was not described.

10/24: "which is circled" – in this figure you also circled values around zero, which do not look like outliers. Please clarify and maybe use two different ways of circling (e.g. circle outliers and use a rectangular box for a zoom).

10/31: "only measured" – I thought you did NOT measure $z_0$, but modeled it. This confusion I have here may relate to the point above: calibration normally requires a standard, but I am not aware of any calibration standards for $z_0$. My best guess is that you made an optimum parameter estimate for $z_0$ in your model, but neigher "measurement" nor "calibration".

11/5: what do you mean with "absence of K flux"? You defined $K$ as the turbulent

exchange coefficient, but here you probably meant "absense of sensible heat flux"?

11/17: "Unless proven otherwise" – there are no proofs in the empirical sciences, thus please reword. According to Popper you can only disprove hypotheses, but not prove them.

11/17-19: this whole sentence is not understandable for me. Please rephrase.

11/26: use "small" in place of "low". And use "substantially" in place of "significantly" - unless you made a statistical test (but then please tell the reader which test and which p value)

11/33: what do you mean with "the model smoothly reproduced"?

Tables: Table captions should be on top of the tables.

Tables 1 and 2: replace the question mark with the correct characters

Table 1: Transmitter should have two t; CS300 should have a range given in $\text{W m}^{-2}$ to which the $\pm 5\%$ information applies

Table 2: same here for SP-Lite and CGR3; m/s should be $\text{m s}^{-1}$

Table 3: $\overline{W}$ is normally the mean vertical wind speed. For horizontal wind speeds, it is more convenient to use $\overline{U}$.

Table 4: You give K, L, H, $U_E$ in $\text{MJ h}^{-1}$. This could be converted to W, but the issue is that this is NOT a flux density. The correct unit would be $\text{W m}^{-2}$. My best guess is that $\text{MJ h}^{-1}$ is a typo and should be $\text{MJ h}^{-1} \text{m}^{-2}$. In any case: double-check and report in $\text{W m}^{-2}$.

$U_E$ is in my view not a commonly used symbol for latent heat flux. Please consider using LE or $\lambda$E instead.

Figure 2: figure captions should explain all items found in the figure. Here we lack the information about RMSE, ME, MAE, and the information about the indices "sim" and

"obs" (the latter simply require a mentioning in parentheses after the respective full words).

Figure 4: why are there no snow depth measurements from the first winter and the two most recent winters on the plot?

Figure 5: explain what SWE means. The percentages are written next to the area showing snowmelt, which is confusing. Move the percentages to evaposublimation (lower part; you could also reverse the arrangement and give snowmelt at the bottom of the graph and put evaposublimation on top of it).

Figure 6: write out Pdf. Is this figure really needed? Could it eventually be produced as a logarithmic plot (maybe as log(x+1))?

Figures 7 and 8: you use symmetric uncertainty bars showing standard deviation. Standard deviation is one of the two parameters of a normal distribution. Are your data really normally distributed? If not then rather give some confidence interval (e.g. 95%, but also 50% would be OK as long as it is clearly described in the caption).

Figures 9 and 10: abbreviations in the plot should be explained in the captions.

**References**

Eugster, W. Merbold, L. (2015) Eddy covariance for quantifying trace gas fluxes from soils. SOIL, 1:187-205, doi:10.5194/soil-1-187-2015

---

## Author Comment (AC1) · 29 Oct 2016

**Reply to Anonymous Referee #1**

*In this document, we include the reviewer's comments in black plain font, and our response embedded in the text, in blue italics. Line numbers refer to those of the version in track changes mode.*

**General comments:**

This work details on evaposublimation rates from snow in the Spanish Sierra Nevada. It is based on direct observations over several years as well as numerical simulations using a snow model that has been trained to match observations. The study highlights the hydrological relevance of evaposublimation in meteorological conditions prevalent in the study area, in particular considering its frequency of occurrence.

Although there are quite a few studies that have reported on evaposublimation rates from other parts of the world, I do appreciate the effort put into these measurements. Its combined evaluation in conjunction with the model simulations is generally solid. Calibrating the snow model using fluxes rather than states (but using the latter for validation) makes a welcome component. A missed opportunity is that there appears to be no systematic data available on snow surface characteristics that could have been compared to zo values presented in Table 4. Nevertheless this study should in my opinion be published after addressing the specific comments listed below.

*We are really grateful to anonymous referee #1 for the time and effort spent reading and correcting this manuscript. Thank you for the positive comments. We fully agree with the comment about the lack of data on snow surface characteristics, and, in fact, this is something that will be considered in the design of the future fieldwork experiments.*

**Specific comments:**

(2/26) Shouldn't "latter" be "former", if you include sublimation losses from snow intercepted in forest canopy?

*Well, it is not a trivial question, indeed. It is true that sublimation losses from the intercepted snow can be small or huge, depending on the kind of trees and their density, snow density, total precipitation, temperature regime ... But, on the other hand, in unforested areas we expect stronger winds and we can also find sublimation from the blowing snow. It all depends on several factors for each particular environment, and how some processes dominate over the rest of them. This complexity makes it unfortunate trying to explain in a short sentence that, in fact, does not contribute significantly to the idea of the paragraph. So, following this, we have removed it in the new versions to avoid excessive simplifications (2/32).*

(3/11) Why should this be a problem of the device? A snow lysimeter is to measure snowpack runoff, not snowmelt rates.

*We fully agree. It is not a problem of the device, but a problem for the modeller, who wants to use these runoff data as if they were snowmelt for testing energy balance models. We have*

*rephrased the text on snow lysimeters to be more precise in the statements. Please, see the next comment.*

(3/25) You highlight the simplicity and low costs for traditional manual measurements versus the need for constant maintenance of automatic devices such as snow pillows. But isn't it the manual measurements that require constant maintenance?

*Of course it is; we were in fact thinking about the complexity of maintaining as much as possible non-disturbed conditions, but the phrasing leads to different conclusion. We have changed the text about the lysimeters and the snow pillows together with the previous comment (3-16-24):*

*"1) Snow water equivalent sensors (Jonhson and Marks, 2004) and snowmelt lysimeters with snowpillows (Tekeli et al., 2005) are used in conjunction with the methodology developed for studying evapotranspiration on agricultural lands. Snow lysimeters are a suitable field method for estimating the permeability of a snowpack (Datt et al, 2010). The main problem is that the snow conditions above these automatic devices may differ from those in the natural snow because of the disturbance of the snow-ground interface (Dingman, 2002) or the appearance of snow bridging. Besides, there is a poor correspondence between the meltwater produced at the snow surface and the water arriving at the base of the snowpack on a unit-area basis, which is a problem when we want to test snow energy balance models (Kattelmann, 2000). Moreover, in semiarid mountain areas, the snow pillow measurements are adversely influenced by the typically shallow snow cover and the frequently high wind speed (Schulz and de Jong, 2004)."*

*We have also depicted a less idealistic picture about the pan method (4/9-11):*

*"The main disadvantage of this method is that it provides us with discrete results that have to be obtained manually, and, with respect to EC, that it needs some adequate measures or estimates of the parameters used for calculating the turbulent exchange of latent and sensible heat"*

(5/14) Please mention if any of your data were affected by snow-vegetation interactions.

*Following this, we have included some wording: "always on sites without snow-vegetation interactions" (5/17)*

(5/15) Section 3.1 is a bit lengthy to my taste, given that this is a published model; irrespective of modifications that should of course be described here or in the appendix.

*We have cleaned up this section (6-12-7/2) and moved to Appendix A some of the model details with less influence on the main goals of this work (17/14-25). We have maintained the considerations about the calibration parameters.*

(5/31) Will probably be handled by the type editor, but I suggest to refer to "Appendix A".

*Corrected at 6/15 and 7/1*

(6/16) You may want to add here that L_down is available for simulations at Poqueira, but not for Pradollano.

*This was already outlined in the later section "Meteorological data during field surveys". We have changed the text there to emphasize this difference (9/25).*

(6/35) But z0 is not constant for simulations presented in Table 4.

*It is constant if we consider that each data set test is an independent simulation from the rest of tests in the table. This allows us to calculate a mean z0 that is used, as constant, for the continuous simulation from 2008 to 2015, presented in section 4.3 and in Fig. 3 to 9.*

(7/16) In the abstract you mention 15 field campaigns, here it is 10. Find a consistent wording to discriminate between campaigns (10) and data sets (15).

*This misleading wording has now been corrected throughout the manuscript. Now we only have references to the 10 field campaigns and the 15 data sets/meteorological states.*

(7/22) You use the term evaposublimation referring to evaporation and/or sublimation, but for the revers process only the term condensation is used without specifically addressing possible deposition (resublimation). I suggest using consistent terminology throughout the manuscript (see also at 14/2)

*Yes, this is true and we have revised this wording throughout the paper. Deposition and condensation are mentioned together as "deposition/condensation", in many occasions throughout the text, actually. In correspondence, however, we dare not to "coin" a word like "condeposition" or "depocondensation".*

(7/30) Consider adding a photo of your device. Why/how would the lower tray inhibit further evaporation?

*The lower tray, once the upper one is on it, forms a closed container that prevents meltwater from evaporating out of it.*

*We have added some explanatory photographs of the device and its handling (27/Fig. 2)*

(8/10) Accuracy is more relevant than precision.

*Following this, we have corrected this in the revised text (8/31)*

(8/28) "Surroundings" is a bit unspecific, within how many meters of the station?

*We have rephrased the sentence as follows: "The tests on the southern face of Sierra Nevada were carried out in an area within a radius of 20 m from the permanent weather station at the Refugio Poqueira monitoring site..." (9/14)*

(9/1) Did you account for different instrument heights when modelling N/S sites?

*Yes, we did. The height of the anemometer, za in Eqs. (A5) and (A6), is an input parameter to the model.*

(9/21) It seems contradictory to name E measurements more reliable if you have to omit 2 of those values over 1 omitted M value.

*Yes, it is true, and "reliable" is not the proper word to be used here. Both of the rejected measures of E were due to handling errors that were apparent at a glance as they involved a visible mass exchange between the tray system and its placement site (because of the wind or some accident). What we meant but did not state clearly was that once the experimental work for a given test has been successfully completed, the measurements of E are more prone to be correct than those of M, which depend on a clean drainage through the disturbed bottom surface of the snow. To clarify this, we have rephrased the paragraph (10/11-15):*

*"The melting measurement relies on the correct drainage from the upper tray, which may sometimes be incomplete. We paid special attention to avoid the refreezing of meltwater in the drain holes, which was not observed in any of the performed tests. Three observations, one related to M in test 9, and two other related to E in tests 5 and 7, had to be rejected because they presented measurement errors due to accidents during the experimental work."*

(9/24) Better to provide specific reasoning to delete those values from your results. Outlier removal is a sensitive matter.

*This has been addressed in the answer to the previous comment.*

(10/1) Reword "quasi-constant", this process is not quasi-constant, it may occur most of the time. Moreover, "60% of the time" doesn't seem to match "almost always".

*We agree. It has been change to "On the contrary, evaposublimation is a continuous phenomenon, albeit at low rates." (10/22)*

(10/15) "favorable" for what, for evaposublimation? Then you should probably mention wind.

*Yes, we did not mention this. We have rewritten this as: "...favourable weather conditions for evaposublimation (cold days with low relative humidity and gentle wind speeds around 5.0 m s-1)" (11/3)*

(10/20) Delete "in agreement with the actual conditions" or provide data, in particular if you have some!

*Following this, we have removed it in the revised text (11/8)*

(10/34) "correctly"? Moreover, did you allow K_H0 to vary between experiments or did you force K_H0 to be constant?

*This term has been replaced by "adequately". (11/22)*

*With respect to K_H0, the sensitivity analysis showed that the model is much more responsive for changes in z0 than in K_H0. Moreover, the initially calibrated value for K_H0 was always close to 1W m-2 K-1, a value quite often found in other works in the literature. So in the final simulations we fixed this value.*

(11/8) The three highest errors in E stem from these 2 two periods, so I do not necessarily agree with this statement.

*As we explain in the following paragraph in the manuscript, the error for test 8b should not be considered since it is not due to the calibrated value of z0 or K_H0, but rather to some problem in the modelling of the deposition/condensation process. No valid combination of values for these parameters was capable to simulate the measured deposition/condensation amount.*

*As for the error in test 10a, it accounts for 17% of the measured evaposublimation rate in this test, which reaches 0.110 mm h-1, a value not that high.*

*Finally, the error in 10b, is undoubtedly large (68%).*

*On the other hand, there are 4 tests left, 3 of them with moderate evaposublimation rates and with low error values, and some of them associated to very different meteorological conditions. Moreover, tests 10.X also involved melting, which is simulated with low error values.*

*Taking all this into consideration, we think this statement can be maintained as it is. (11/29-32)*

(12/1) The order of statements seems strange. Given that you used flux data to calibrate your model, it should primarily replicate the flux terms, and eventually also the states, not the other way around.

*It is correct. But as in this section we are talking about the validation, which is tested against the observations of snow depth, it makes sense to express it in this order. Besides, the reference allows us to show that the good representation of the timing in the snow cycles supports the conclusion about the calibrated fluxes remaining well simulated during validation. (12/27-29)*

(12/34) Just out of curiosity, is G too small / irrelevant to be shown?

*We have considered it negligible in the modelling.*

(13/25) Please restrict this statement to your field site or the meteorological conditions in the Sierra Nevada. Moreover, from Table 3 it seems there are 3 instances of observed zero or negative E.

*Following this comment, we have changed the sentence as follows: "The measurements confirm that, for the study sites in Sierra Nevada, the evaposublimation rate is…" (14/20)*

*As for the comment on the observed zero values, we meant zero for evaposublimation or deposition/condensation, not only evaposublimation. This is corrected in the revised version. So, in Table 3 we can find 2 instances of observed zero values. But the value of E in test 8a is a false 0 (please see (10/23-30 and 11/33-34),, as it is in fact a sequence of evaposublimation followed by a deposition/condensation equivalent in magnitude. The only real zero value appears in test 10b.*

*After the Reference's comment, this sentence is redefined: "Only in one (10b) of the 15 measured meteorological states did evaposublimation or deposition/condensation appear to be inhibited:..." (14/22)*

(14/2) What/where is "in the Alpine summer"?

*It is a bizarre way of saying: "in the Alps during the summer". Corrected. (14/30)*

(14/14) What do you mean by "high resolution"? I don't agree with your reasoning. You observed condensation at Poqueira, this is where you do have local meteorological data. Speaking of surface hoar formation and associated mass fluxes: you may want to look at a Stössel et al (2010, doi:10.1029/2009WR008198).

*Thank you for the reference. This is something we would like to further investigate in future research, so we will use it.*

*We meant high spatial resolution, that is, <10 m according to Feick et al (2007). Our sentence is certainly misleading so we have changed it as follows (15/8-11):*

*"The simulation of hoar growth in complex terrain is a difficult task since it demands data of the local wind regime with a spatial resolution under 10 m (Feick et al, 2007), which was not accomplisheded for the tests in Poqueira, located 10 to 20 m away from the station."*

(22/Table 1) This table is incomplete, please add a header and remove misprinted characters such as the "?".

(23/Table 2) This table is incomplete, please add a header and remove misprinted characters such as the "?".

*Both have been added/removed in the revised text (24/Tables 1 and 2)*

(23/Table 3) "Solar radiation" seems not the best term for what is presented in the column below.

*We agree. We have changed the term by "sky condition" (25/Table 3)*

(25/Figure 2) The third panel on the right could be removed.

(26/Figure 3) The third panel on the right could be removed.

*Both have been removed.(28/Figs 3 and 4)*

(27/Figure 4) Remove 2008/09 data, which is not a complete winter season, also considering that manual measurements commenced in 2009/10.

*Actually, 2008/09 is a complete snow season. The simulation starts just before the first snow event of the water year, which was also of considerable magnitude. That is the reason why it may seem from the figure that the simulation was started with an initial condition of an already accumulated snowpack, but this is not the case.*

*Despite the field tests started in 2009/10, once we obtained a calibrated version of the model from this data, we decided to use all the available data-period at the Poqueira station. 2008 is the starting date for the meteorological measurements at Refugio Poqueira weather station with its present configuration. This allows us to include all the observed variability in the snow regime at this site. In fact, the 2008/09 season was outstanding because of the large amount of accumulated snow and the persistence of the snowpack. (29/Fig 5)*

(28/Figure 5) Combining Figure 5) with Figure 4a) seems to indicate very low snow densities in years such as 2011/12 and 2013/14. Are these values correct?

*We have checked the results and they are correct. The snow densities are always in the expected range, according to the parameterization of the snow density in Appendix A (Eqs (A9) and (A10)). These interannual figures may not be the best way to capture the evolution of the snow density. Besides, in these two years (2011/12 and 2013/14) there was a very poor snow presence with a snowpack that melted systematically and quickly after each snowfall. (29/Fig 5 and 29/Fig 6)*

(29/Figures 7-10) These Figures take quite some space, but there is comparably little text in the body of the paper associated with these figure. Consider deleting two of the figures or extending the associated text.

*We agree with the reviewer. We have removed Figs 7 and 10 and consequently adapted the associated text. The other 3 figures are commented in two complete paragraphs at the end of the section "Results" that we consider important because they describe the mean values and the monthly variability of the mass and energy fluxes.*

---

## Author Comment (AC2) · 29 Oct 2016

**Reply to Referee #2 Werner Eugster**

*In this document, we include the reviewer's comments in black plain font, and our response embedded in the text, in blue italics. Line numbers refer to those of the **version in track changes mode**.*

**General comments:**

The authors modeled snowmelt and evaporation/sublimation losses from a snow pack in the Sierra Nevada, Spain, using a point energy budget model over 7 winter seasons (2008/9 to 2014/15). Their model suggests that 24 to 33% of the annual ablation of the snow pack is not via meltwater, but via gaseous vapor losses ("evaposublimation").

Although I have some critical remarks, I find the paper sound, relevant and suitable for the journal after the necessary revisions.

My own background is rather in eddy covariance flux measurements which are not employed here, and hence some critical remarks relate to the fact that from reading the paper I got the impression that the authors would have really profited from eddy covariance flux measurements, which are no longer as difficult to perform as e.g. Hock (2005) thought more than 10 years ago. There are probably 3 sites in Spain that might have data for follow-up studies:

Castellar de N'Hug, Spain, Pyrenees: https://fluxnet.ornl.gov/site/4055

Lanjaron, Spain, Sierra Nevada: https://fluxnet.ornl.gov/site/4060

Laguna Seca, Spain, Sierra Nevada: https://fluxnet.ornl.gov/site/4058

With eddy covariance one could directly measure sensible heat flux and thus the somewhat weakly justified assumption made by the authors that the turbulent transfer coefficient for sensible heat, $KH0 = 1$ W m$^{-2}$ K$^{-1}$ could have been omitted. Moreover, $z0$ could have been derived from the direct measurement of momentum flux and horizontal wind speed, and the bascially tuned value for $z0$ of 0.61 mm (both given on page 11, line 31) would have led to a more vigorous testing of the model.

If you add latent heat flux measurements, then of course eddy covariance flux measurements become more demanding, but already a simple sonic anemometer would provide the information mentioned above.

Having said so, I still think that the authors did a good job with the approach they used and I hope that addressing the following major points helps to improve the paper before final acceptance.

*We thank Prof. Dr. Eugster for his reading of our paper and the insights into the using of eddy covariance techniques for the study of the evaposublimation presented here. Also the rigor with which he has addressed the treatment of the boundary layer theory (and the rest of the concepts in the paper in general) is very much welcome and appreciated.*

*We are not experts in that field of eddy covariance, but we agree with his suggestion of following up this analysis of evaposublimation with some detailed data obtained with eddy covariance measurements. In fact, we have an ongoing study together with researchers working in the two fluxnet sites located in Sierra Nevada, Lanjarón and Laguna Seca, and we had even done some preliminary analysis over their EC data. These sites could not be used for the current work, however, due to different reasons. Firstly, their location was not selected to perform specifically measurements over snow but rather of different processes. The Lanjarón site is located at an altitude of approximately 2250 m and this EC system was operative during 2009. It consisted of two EC towers, one of which was located over burnt pines that were left standing for the selected post-fire treatment, while the other was only operative from June to December. The Laguna Seca site was located at an elevation of 2267 m and its EC system was operative for two years (2007-2008). Despite we have access to some useful measurements over snow of roughness and latent heat flux, this site is located quite far away from the two monitoring sites for snow in this paper. Secondly, since these two sites were not intended to study the snow, these stations were not provided with specific sensors for snow monitoring, like snow depth, rain gauge, or camera. The distance between both groups of dataset makes it complex to use directly these EC measurements to derive conclusions in our work without further and rigorous analysis, and without additional measurements over these sites. That is the reason why we have not used them in the present work.*

*Nonetheless, we fully agree that the use of eddy covariance data will derive a sound validation/contrast of the estimations of this work and its conclusions, and it is a desirable further step in our future design of field and experimental work. In this sense, we also note the interesting article by Eugster & Merbold (2015) that will be of help in the future design of these improved experiments to measure evaposublimation from the snow at these mountainous sites. Likewise, in these future experiments the measurement of z0 will be a priority, for sure, even when EC sensors were not available.*

Major Points

1. 2/21-22: "The evaposublimation rate depends on the vapour pressure gradient between the surface of the snow and the air, which is mainly influenced by the local wind intensity, and hence, by the complex turbulent phenomena occurring in the boundary layer." – I think you should more clearly phrase that in the first place you need a lot of energy to evaporate or sublimate water. It is not primarily the vapor pressure gradient that drives the flux, it is the heat supply to the snow surface (which is of course related to all gradients). Please rephrase. Actually on line 30 on same page you have a remark about solar radiation, but not a general picture of the relevance of energy fluxes.

*We cannot but agree with this comment; reading this piece of text, it is clear that the unfortunate selection of words has made this sentence mean something different from what it was intended for. To convey the referee's comments to the paper, we have rephrased these initial sentences in the paragraph (2/22-28):*

*"One of the mass balance fluxes in the snowpack is the water vapour exchange between the snow surface and the atmosphere. It is directly linked to the latent heat flux and it is governed by the complex turbulent phenomena occurring in the boundary layer. The evaposublimation*

*process requires a high amount of energy available at the snowpack to complete the phase transition (e.g., Strasser et al., 2008). The evaposublimation rate can be calculated as a function of the vapour pressure gradient between the surface of the snow and the air, and it is decisively influenced by the local wind intensity and turbulence."*

2. 3/21-22: You mention that eddy covariance flux measurements "are complex, fragile, and require large, clear, low–angle areas to function optimally". This is not really correct. The measurements actually function quite nicely, but the key issue is that they are point measurements, and although they might be very accurate point measurements, the relation of these point measurements to the surface area is a challenge. See point 3 below. As an example: why do large eddy simulation (LES) people simply use arrays of ultrasonic anemometers to obtain field data? Because they can use exactly such point measurements to validate their models.

*These comments about the fragility of EC sensors, based on others' experiences, were meant to be ascribed to EC over snow under high mountain climatology. After the reviewer's comments, we have changed and added several sentences throughout the paper to depict a more realistic and updated state of the art regarding the Eddy Covariance (EC) techniques. We have also included some text to show the advantages that EC would bring to this study, especially regarding the omission of the two calibration parameters, as pointed out by the Referee.*

*Changes in the introduction (3/30-4/1 and 4/9-11):*

*"EC instrumentation is quickly evolving during the last years, and successful applications under a wide variety of environments can already be found (e.g. Reverter et al 2010, Eugster and Merbold, 2015, Knowles et al, 2015), as it is no longer as complex and fragile as it used to be. EC provides very accurate point measurements, even though the translation of these point data to a surface area still represents a challenge nowadays (Eugster and Merbold, 2015). However, experiments using EC systems are still expensive and time consuming, as the data obtained demand complex and rigorous analysis with corrections and post--processing to ensure measurement accuracy (Reba et al., 2009)."*

*"The main disadvantage of this method "(evaporation pan) "is that it provides us with discrete results that have to be obtained manually, and, with respect to EC, that it needs some adequate measures or estimates of the parameters used for calculating the turbulent exchange of latent and sensible heat"*

3. 13/8-10: The authors write "The validity of the application of boundary layer theory to determine the turbulent fluxes over the snow, especially on complex mountainous terrains, is not clear (Hock, 2005)" – which sounds quite special. I double-checked the Hock (2005) reference and thus do not think that this wording can withstand a careful check. First, several statements in Hock (2005) are by now outdate, e.g. the idea that the eddy covariance technique "require[s] sophisticated instrumentation with continuous maintenance, which render them unsuitable for operational purposes. Consequently, such studies are rare and restricted to short periods of time". In the meantime there are many such continuous measurements. Second, this means that all other statements are focusing on the flux-gradient

method that she is interested in, thus it cannot be deduced that her judgment applies to all possible approaches. Third, the fact that some authors "lack an explanation in terms of boundary layer theory" (page 378, left column) does not mean that such a theory is not valid.

What would be an acceptable summarization of the Hock (2005) paper in this context could be written e.g. with the wording "Flux-profile and bulk transfer approaches have been shown to be problematic over sloping terrain to determine turbulent fluxes (Hock, 2005)". It is essential to make clear that it is not a problem of the theory, but of the flux-gradient or bulk method that Hock (2005) talks about. In principle one could use eddy covariance, but also there could be issues since this is a point measurement and the relation to the footprint area influencing that measurements is challenging (if you need a more detailed explanation then please consult Eugster and Merbold, 2015).

You use exactly that boundary-layer theory in your modeling approach (e.g. equations A3 and A4) and there you found a more appropriate wording for summarizing the information given by Hock (2005).

*This was, in fact, a clear misunderstanding of Hock (2005) and a lack a rigor on our part when summarizing her work. We thank the Referee for pointing out this inconsistency. Following his remark, we have changed this piece of text accordingly (14/2-5):*

*"The turbulent heat transfer terms are probably the most uncertain contribution to solving the energy budget over the snow. Flux--profile and bulk transfer approaches have been shown to be problematic over sloping terrain to determine turbulent fluxes (Hock, 2005)"*

4. Data availability: it would be great if the data could be placed in a long-term archive, such as www.pangaea.de (which is free of charge for the authors)

*This is an interesting suggestion that we have followed. Data submission is in process and they will be soon available at www.pangaea.de.*

**Details**

1/9 and many more places: "m.a.s.l." has one period too much: there is never a period after m for meters. Thus "m a.s.l."

*This has been revised throughout the text*

1/10: "The ratio is changeable" – do you mean "variable"? or what does this actually mean?

*Please, see next comment.*

1/11: "timing of the meteorological inputs, generally unforeseeable in this semiarid region" – there is something wrong here. I do not really get what you want to express. The timing of INPUT sounds incorrect in this context, and thus I do also not understand the connection with "unforeseeable" (maybe you mean that forcasting weather conditions does not work in this semiarid region? But are you sure there is no skill at all in such forecasts?) - please rephrase.

*We apologize for this phrasing. We tried to emphasize the highly variability that both rainfall and snowfall regimes exhibit here in southeastern Spain, which has a major influence on the*

*snow persistence and metamorphosis during the cold season. This variability causes that the starting and ending date of the snow in a given water year, its duration, and the maximum accumulated snow water equivalent, among other variables, change hugely between consecutive years, for example. We have rephrased the sentence to clarify it (1/11):*

*""This ratio is very variable throughout the year and between years, depending on the particular occurrence of snowfall and mild weather events, which is generally quite erratic in this semiarid region."*

1/16: "as the latitude descends" – please rephrase, the latitudes stay in place. My understand is that you wanted to say "at increasing altitudes with decreasing latidude".

*This sentence has been changed following this comment (1/17)*

2/13: only use the word "significant" in the context of statistical significance tests. If it does not relate to statistics, then use other words that do not have a special meaning in scientific texts. But here there is an error: "significant data series" does not sound correct anyway. Maybe you wanted to say something about data availability (no gaps, long time series?)

*Yes, in this case the Referee has mentioned, we meant "continuous". Following this comment, we have replaced "significant" throughout the manuscript with more precise terms.*

2/15: "source of distributed data" – probably "source of spatial data"?

*Following this comment, we have replaced this word in the revised version (2/17).*

2/20: what do you mean with "latent heat balance"? Probably "latent heat FLUX"?

*Yes, this was an editing mistake. We have corrected this word in the revised version (2/24).*

5/3: what is an "alter-shielded rain gauge"? Please explain in more detail or give a reference where I could inform myself about this term that I do not know.

*This is a term commonly found in the bibliography (eg. Fassnacht, 2004) given to a rain gauge equipped with an alter shield (Alter, J. C. 1937. Shielded storage precipitation gauges. Mon. Wea. Rev. 65. 262265) "to improve snow catch in windy conditions". We have added this explanation and the interesting reference of Alter (1937) about the original design to the text (5/18).*

5/20 and many places elsewhere: you seem to have had some trouble with the characterset and all these question marks most likely should have a specific meaning. Please search for all question marks in the text and make sure that in the revisions you get the correct characters everywhere.

*It was a typo error related to the hyphens present in 5/20 and in tables 1 and 2.*

6/4: do not use computer code writing in text passages. Here you should use ≤

*This has been corrected (17/15).*

6/19: add "Appendix" before A

*This has been added in the revised version (6/15 and 7/1).*

6/25: you cite Calanca (2001), but he does not primarily look at the aerodynamic roughness length, but at the roughness length for temperature. This topic again is related to the issue that you did NOT use eddy covariance flux measurements. With EC flux measurements you would bypass this issue. The information that the Calanca (2001) reference relates to actually would rather fit the information on line 34, same page. Best would be to rewrite and make sure the confusion between aerodynamic roughness (z0) and roughness length for temperature (θ0) is resolved.

*Calanca (2001) measures both z0 and zT (your θ0) and he reaches in his work some conclusions valid for both of them and for their ratio; that is the reason why we used his work as a reference in the text. However, following this comment, we could identify some misleading use on our side of the different terms involved in the concept of roughness, and we have modified this paragraph accordingly (7/6 and 7/16-17).*

7/11-12: "According to Braithwaite (1995), uncertainty in z0 may cause larger errors than neglecting stability." – this actually is a strong argument why you should try out eddy covariance in follow-up research! With EC you measure both z0 and z/L (or better: you can compute these two from the raw measurements).

*We fully agree with the Referee and our future steps will for sure follow the EC measurements approach.*

8/22 and Tables 1 and 2: here is an error: the ±5% uncertainty does not relate to the range of wavelengths that the sensor is sensitive to, but to the units of measurements, which are W m−2 . All other sensors except radiation sensors in the tables have range of measurement in correct units ± uncertainty, please give the same for radiation sensors and specify their wavelength sensitivity elsewhere (e.g. for snow temperature you mention 2 levels, you could do the same for solar radiation and write 300–1100 nm in parentheses).

*We thank the Referee for this comment; we have corrected this mistake in Tables 1 and 2 and (9/8).*

9/2: "The air vapour pressure was determined by the standard psychrometric method." – I am not convinced about this: the standard psychrometric method uses a dry bulb and wet bulb temperature sensor. You however do not mention a wet bulb sensor, but a relative humidity sensor on page 8, last line. Thus, you calculated vapour pressure differently - please correctly inform us how you calculated it from temperature and relative humidity (most likely you used some equation like the Magnus equation to determine saturation vapor pressure at air temperature, then used relative humidity to calculate actual vapor pressure).

*The Referee is absolutely right, and this was a mistake on our side. We actually use the empirical equation in Dingman (2002), which has the same mathematical form as the Magnus Tetens formula and differs only in the parameters. This information has been corrected in the text (9/22):*

*"The air vapour pressure was calculated from T_a and RH_a using the empirical relation in Dingman (2002)."*

9/2: "using a standard pyranometer in both cases" – I do not agree. In Tables 1 and 2 you show that you use a CS300 at one site and an SP-Lite at the other. Both are silicon photovoltaic detector sensors, that are calibrated against a standard pyranometer, but they are NOT standard pyranometers! Please reword. For more information: https://www.campbellsci.com/cs300-pyranometer, http://www.kippzonen.com/Product/9/SP-Lite2-Pyranometer

*Again, we apologize for this misleading wording. The use of "standard" is quite unfortunate, and we really meant "common" pyranometers. We have replaced this adjective by the more precise " silicon photovoltaic" (9/23).*

9/27: "Moreover, temperature was found to be a necessary but not sufficient driver for melting." – In fact, it is the sensible heat flux, which is a function of the temperature GRADIENT. Please be more precise in your wording.

*Following this comment, we have changed the sentence (10/17):*

*"Moreover, temperature was found to be a necessary but not sufficient cause for melting."*

9/28 and elsewhere: you are not consistent in how you print physical units such as ◦C or mm h−1 , sometimes in italics, sometimes not. Please homogenize (see the guidelines)

*We have revised the text and homogenized the format of units accordingly to the guidelines of the journal.*

9/29: "positive heat input from shortwave radiation" – this is another shortcut that students tend to misunderstand. Please reword and make sure it is clear that shortwave radiation is a high-level form of energy which first needs to dissipate to heat, but shortwave radiation by itself is NOT a heat input.

*Following this comment, we have rewritten the sentence as "positive heat input caused by the dissipation of shortwave radiation" (10/19)*

10/1: what is meant with "quasi-constant"? is it "continuous" (in opposition to sporadic)?

*Yes, this was what we meant; we have replaced this term by "continuous" (10/22).*

10/20: how did this calibration go for z0? This was not described.

*Following this comment, the following explanation has been added to the text: z0 "was estimated by minimizing the sum of the mean errors in E and M". This explanation was added to the text (11/9). See also the answer to comment 10/31.*

10/24: "which is circled" – in this figure you also circled values around zero, which do not look like outliers. Please clarify and maybe use two different ways of circling (e.g. circle outliers and use a rectangular box for a zoom).

*The circle is highlighting the same test in the three panels of Fig. 2. This test appears as an outlier only in the left panel, what means that it is an outlier for E, but not for M and E/(E+M), as melting is not observed nor simulated.*

*Following the other Reviewer, the right panel if Fig. 2 has been removed*

10/31: "only measured" – I thought you did NOT measure z0, but modeled it. This confusion I have here may relate to the point above: calibration normally requires a standard, but I am not aware of any calibration standards for z0. My best guess is that you made an optimum parameter estimate for z0 in your model, but neither "measurement" nor "calibration".

*Yes, "measured" is not the correct word. We have changed it to "estimated" (11/20). And we have also replaced "calibrated" in comment 10/20 to "estimated" (11/8).*

11/5: what do you mean with "absence of K flux"? You defined K as the turbulent exchange coefficient, but here you probably meant "absence of sensible heat flux"?

*K refers to the shortwave radiation (Eq (A2)), while K followed by a subindex refers to the respective turbulent exchange coefficients associated to each energy flux in the balance equation. Following this, to avoid confusion, we have rewritten the sentence: "as the absence of shortwave radiation (K flux in Eq. A2)"..." allows us to better adjust the calibration parameters in the energy balance" (11/28).*

11/17: "Unless proven otherwise" – there are no proofs in the empirical sciences, thus please reword. According to Popper you can only disprove hypotheses, but not prove them.

*This expression is wrong and, in fact, unnecessary. We have removed it (12/6).*

11/17-19: this whole sentence is not understandable for me. Please rephrase.

*We have rewritten the whole sentence as: "This difference in the deposition/condensation rate is not likely to be due to a measurement error but to a modelling issue. The model succeeded in reproducing the sequence of deposition/condensation and sublimation but missed the deposition/condensation rate by an order of magnitude. Further work is needed to test this deviation by the model and identify its sources." (12/7-11)*

11/26: use "small" in place of "low". And use "substantially" in place of "significantly" - unless you made a statistical test (but then please tell the reader which test and which p value)

*Following this comment, we have replaced both terms in the text (12/17 and 12/19).*

11/33: what do you mean with "the model smoothly reproduced"?

*We mean that the model reproduced the patterns adequately, without strong shifts. After this comment, we have simply removed this adverb (12/26).*

Tables: Table captions should be on top of the tables.

*It is right. This was corrected, together with the figure captions, which were on top when they should be under the figure, according to the journal editing guide.*

Tables 1 and 2: replace the question mark with the correct characters

*As stated in a previous comment, this has been corrected throughout the text.*

Table 1: Transmitter should have two t; CS300 should have a range given in W m−2 to which the ±5% information applies

*This has been corrected, also in agreement with comment 8/22 (24/Table 1).*

Table 2: same here for SP-Lite and CGR3; m/s should be m s−1

*Done (24/Table 2)*

Table 3: W is normally the mean vertical wind speed. For horizontal wind speeds, it is more convenient to use U.

*Changed (25/Table 3). Moreover, we have noticed that W was already in use as the mass transport due to wind in Eq (A1)*

Table 4: You give K, L, H, UE in MJ h−1 . This could be converted to W, but the issue is that this is NOT a flux density. The correct unit would be W m−2 . My best guess is that MJ h−1 is a typo and should be MJ h−1 m−2 . In any case: double-check and report in W m−2 .

*Yes, this is a typo: the "m-2" were missing on the table. The model uses MJ h-1 internally, and we missed to change the units to W in the Table, as we did in Figs. 9 and 10. This has been corrected (26/Table 4).*

UE is in my view not a commonly used symbol for latent heat flux. Please consider using LE or λE instead.

*Despite not being the most commonly used notation, the adoption of UE as the product E.uE is consistent and highlights the fact that the unitary internal energy of water at a given state may result from different antecedent processes (i.e., warming/cooling and/or change of phase); this is interesting when evaporation and sublimation (or their reverse process) may alternatively or simultaneously occur. However, both suggestions from the Referee are actually much more frequent in literature; following this comment, we have replaced U_E by LE, together with K_UE, now K_LE, throughout the document.*

Figure 2: figure captions should explain all items found in the figure. Here we lack the information about RMSE, ME, MAE, and the information about the indices "sim" and "obs" (the latter simply require a mentioning in parentheses after the respective full words).

*Following this comment, we have added the whole information when needed. (28/Fig. 3, 28/Fig 4, 29/Fig 5 and Table 2).*

Figure 4: why are there no snow depth measurements from the first winter and the two most recent winters on the plot?

*2009-2013 was the period with snow depth measurements available for this study. (29/Fig 5)*

Figure 5: explain what SWE means. The percentages are written next to the area showing snowmelt, which is confusing. Move the percentages to evaposublimation (lower part; you could also reverse the arrangement and give snowmelt at the bottom of the graph and put evaposublimation on top of it).

*Following this comment, we have moved the text with the percentage values to the bottom of the graph and written out SWE (29/Fig 5).*

Figure 6: write out Pdf. Is this figure really needed? Could it eventually be produced as a logarithmic plot (maybe as log(x+1))?

*We have written out Pdf (Fig 6). With this figure we bring attention onto the different occurrence of evaposublimation and melting fluxes, and the comparison of their respective order of magnitude We did try the logarithmic version, but the resultant plot did not improve much the visualization. (30/Fig 8)*

Figures 7 and 8: you use symmetric uncertainty bars showing standard deviation. Standard deviation is one of the two parameters of a normal distribution. Are your data really normally distributed? If not then rather give some confidence interval (e.g. 95%, but also 50% would be OK as long as it is clearly described in the caption).

*We just wanted to show the value of the standard deviation for each set of simulated monthly values (that is, 7 values in each set) in the graph, to highlight the annual variability that is observed in this area, which the model captures. We are aware that the data are too few to adjust a function or to obtain a confidence interval, and that is the reason why no further analysis was performed*

Figures 9 and 10: abbreviations in the plot should be explained in the captions.

*They have been explained in the captions in all cases (31/Fig 10)*

References

Eugster, W. Merbold, L. (2015) Eddy covariance for quantifying trace gas fluxes from soils. SOIL, 1:187-205, doi:10.5194/soil-1-187-2015